

# The GESAMP atmospheric iron deposition model intercomparison study

Stelios Myriokefalitakis[1], Akinori Ito[2], Maria Kanakidou[3], Athanasios Nenes[4], Maarten C. Krol[1], Natalie M. Mahowald[5], Rachel A. Scanza[5], Douglas S. Hamilton[5], Matthew S. Johnson[6], Nicholas Meskhidze[7], Jasper F. Kok[8], Cecile Guieu[9], Alex R. Baker[10], Timothy D. Jickells[10], Manmohan M. Sarin[11], Srinivas Bikkina[11], Morgane M. G. Perron[12] and Robert A. Duce[13]

[1] Institute for Marine and Atmospheric Research (IMAU), Utrecht University, 3584 CC Utrecht, The Netherlands
[2] Yokohama Institute for Earth Sciences, JAMSTEC, Yokohama, Kanagawa 236-0001, Japan
[3] Environmental Chemical Processes Laboratory (ECPL), Department of Chemistry, University of Crete, 70013 Heraklion, Greece
[4] School of Earth & Atmospheric Sciences, Georgia Institute of Technology, Atlanta, GA 30332
[5] Department of Earth and Atmospheric Sciences, Cornell University, Ithaca NY, 14853, USA
[6] Earth Science Division, NASA Ames Research Center, Moffett Field, CA, USA
[7] Marine, Earth, and Atmospheric Sciences, North Carolina State University, Raleigh, NC, USA
[8] Department of Atmospheric and Oceanic Sciences, University of California, Los Angeles, CA 90095, USA
[9] Laboratoire d'Océanographie de Villefranche (LOV), UMR7093, CNRS-INSU-Université Paris 6, Villefranche sur Mer, France, France
[10] Centre for Ocean and Atmospheric Sciences, School of Environmental Sciences, University of East Anglia, Norwich, UK
[11] Geosciences Division, Physical Research Laboratory, Ahmedabad, India
[12] Institute for Marine and Antarctic Studies, University of Tasmania, Hobart, Tasmania, Australia
[13] Departments of Oceanography and Atmospheric Sciences, Texas A&M University, College Station, TX, USA

*Correspondence to*: Stelios Myriokefalitakis (s.myriok@uu.nl) and Akinori Ito (akinorii@jamstec.go.jp)

**Abstract.**

This work reports on the current status of global modelling of iron (Fe) deposition fluxes and atmospheric concentrations and analyses of the differences between models, as well as between models and observations. A total of four global 3-D chemistry-transport (CTMs) and general circulation (GCMs) models have participated in this intercomparison, in the framework of the United Nations Joint Group of Experts on the Scientific Aspects of Marine Environmental Protection (GESAMP) Working Group 38, "The Atmospheric Input of Chemicals to the Ocean". The global total Fe (TFe) emissions strength in the models is equal to ~72 Tg-Fe yr$^{-1}$ (38–134 Tg-Fe yr$^{-1}$) from mineral dust sources and around 2.1 Tg-Fe yr$^{-1}$ (1.8–2.7 Tg-Fe yr$^{-1}$) from combustion processes (sum of anthropogenic combustion/biomass burning and wildfires). The mean global labile Fe (LFe) source strength in the models, considering both the primary emissions and the atmospheric processing, is calculated to be 0.7 (±0.3) Tg-Fe yr$^{-1}$, accounting for mineral dust and combustion aerosols together. The multi model ensemble global TFe



and LFe deposition fluxes into the global ocean are calculated to be ~ 15 Tg-Fe yr$^{-1}$ and ~0.3 Tg-Fe yr$^{-1}$, respectively.

The model intercomparison analysis indicates that the representation of the atmospheric Fe cycle varies among models, in terms of both the magnitude of natural and combustion Fe emissions as well as the complexity of

5    atmospheric processing parametrizations of Fe-containing aerosols. The model comparison with aerosol Fe observations over oceanic regions indicate that most models overestimate surface level TFe mass concentrations near the dust source regions and tend to underestimate the low concentrations observed in remote ocean regions. All models are able to simulate the tendency of higher Fe loading near and downwind from the dust source regions, with the mean normalized bias for the Northern Hemisphere (~14), larger than the Southern Hemisphere

10   (~2.4) for the ensemble model mean. This model intercomparison and model–observation comparison study reveals two critical issues in LFe simulations that require further exploration: 1) the Fe-containing aerosol size distribution and 2) the relative contribution of dust and combustion sources of Fe to labile Fe in atmospheric aerosols over the remote oceanic regions.





## 1 Introduction

Oceans are important for the Earth System's functioning, currently absorbing roughly 27% of total $CO_2$ emissions (e.g., Le Quéré et al., 2013), providing about half of atmospheric oxygen and being a source of biomass needed to help sustain life on our planet. Iron (Fe) is a key element for marine life (Duce and Tindale,

1991; Fung et al., 2000) being required for photosynthesis and respiration. As an essential micronutrient, Fe (co-)limits ocean productivity over large regions (Boyd et al., 2005; Jickells et al., 2005; Martin et al., 1991; Moore et al., 2013), influences the nitrogen fixation capability of diazotrophs in oligotrophic regions (Falkowski, 2000, 1997) and overall affects the transport and sequestration of carbon into the deep ocean (Maher et al., 2010). Atmospheric deposition is considered an important external Fe source to the open ocean (Jickells et al., 2005;

Tagliabue et al., 2017). Micronutrient Fe delivered through atmospheric pathways may influence the primary and export production of carbon over the High-Nutrient Low-Chlorophyll (HNLC) oceanic regions (i.e., the oceanic regions where Fe is the limiting factor for phytoplankton productivity) (Fung et al., 2000; Krishnamurthy et al., 2009).

Understanding of the impact of Fe on global marine productivity requires knowledge of the rates and locations of

Fe supply to the ocean, and of the physicochemical forms of Fe that can be utilized by marine biota (i.e., those that are bioavailable). The bioavailability of Fe is a complex issue (e.g., Lis et al., 2015; Morel et al., 2008) and several naming conventions and abbreviations are used to characterize the atmospheric supply of the bioavailable fraction of Fe to the global ocean (Baker and Croot, 2010; Shi et al., 2012). However, it has been widely assumed that soluble Fe can be considered, as a first approximation, for bioavailable Fe (Baker et al., 2006a, 2006b) and a

common experimental practice to determine the bioavailable Fe portion in Fe-containing aerosols is the quantification of Fe in a leachate solution that passes through 0.45 μm, 0.2 μm or 0.02 μm sized filter (see Meskhidze et al., 2017 and ref. therein). This filterable Fe contains both the soluble Fe and the particulate Fe of diameter smaller than 0.45, 0.2 or 0.02 μm. Upon deposition to the surface ocean, this fraction of Fe from the atmosphere can either enter the dissolved Fe pool, or precipitate-out as large oxy-hydroxide particles (Meskhidze

et al., 2017). Consequently, the impact of atmospheric Fe on marine biogeochemistry depends on both the total Fe (TFe) deposition and its solubility, keeping in mind that the bioavailable fraction of Fe in seawater will then also change due to post-atmospheric deposition ocean processes (e.g., Baker and Croot, 2010; Chen and Siefert, 2004; Meskhidze et al., 2017; Rich and Morel, 1990).

On the global scale mineral dust is the dominant source of Fe to the atmosphere ( ~95%; Mahowald et al., 2009)

The average Fe content in upper crustal minerals is 3.5% (e.g., Duce and Tindale, 1991), but this can vary



considerably, depending on the underlying mineralogy (and geography) of the dust source (Journet et al., 2014; Nickovic et al., 2012, 2013). According to Journet et al. (2014), the Fe content of various minerals is as follows: hematite (69.9%), goethite (62.8%), chlorite (12.3%), vermiculite (6.71%), illite (4.3%), smectite (2.6%), feldspars (0.34%) and kaolinite (0.23%) in clay- and silt-sized soil particles (i.e., soil particles with diameters <

2μm and 2 - 50 μm, respectively).

Fe-containing aerosols also originate from wildfires and biomass burning (Guieu et al., 2005; Mahowald et al., 2005; Oakes et al., 2012; Paris et al., 2010) and anthropogenic combustion processes, such as coal and oil fly ash (Luo et al., 2008), including ship oil combustion (Ito, 2013). Other sources of Fe-oxides can be also be identified in the atmosphere, attributed to volcanic eruptions (Benitez-Nelson et al., 2003; Langmann et al., 2010) and to

lesser extent meteors (Johnson, 2001).

Numerical models are currently the only feasible means to study atmospheric Fe supply to the oceans and the only way to assess its impacts on large (regional and greater) scales in response to changes in past and future climate. However, modelling of the atmospheric supply of soluble Fe to the global ocean is challenging, due to the multitude and complexity of the forms under which Fe can be present in aerosol emitted to the atmosphere

(Meskhidze et al., 2017), as well as the variety and complexity of processes which alter the solubility of Fe during its transport through the atmosphere (Baker and Croot, 2010). Indeed, the soluble fraction of Fe in atmospheric aerosols may include different Fe forms in the ferric oxidation state (Fe(III)) (Fu et al., 2012), ferrihydrite and amorphous precipitates, Fe-oxide nanoparticles (Shi et al., 2009), Fe-organic complexes (Cheize et al., 2012) and Fe in the ferrous oxidation state (Fe(II)) (Raiswell and Canfield, 2012). The atmospheric

modelling community is mostly focused on the soluble fraction of the deposited Fe over the oceans and for this work the general term labile Fe (LFe) will be used to represent the overall soluble Fe in simulated atmospheric aerosol.

During recent decades, intensive research has been carried out to elucidate the origin, nature and magnitude of LFe fluxes to the surface ocean. Soils may include a small fraction of LFe - roughly 0.1%; e.g., Ito and Shi

(2016) - considered as impurities attached on minerals such as illite, smectite, kaolinite and feldspars (e.g., Ito and Xu, 2014). Fe-containing fly ash has been observed to be present as ferric sulfate salts or nanoparticulate Fe and thus is highly soluble (Fu et al., 2012; Schroth et al., 2009), since it is mainly formed via high-temperature combustion followed by sulfuric acid condensation (Sippula et al., 2009). However, the form and the chemical properties of Fe in emissions can vary substantially for each combustion source (Ito, 2013; Wang et al., 2015),

with the initial soluble fraction in combustion emissions to be about 77–81% in oil fly ash (Schroth et al., 2009), 20–25% in coal fly ash (Chen et al., 2012) and 18-46% in biomass fly ash (Bowie et al., 2009; Oakes et al.,




2012). Recently, Matsui et al. (2018) suggested, based on observed magnetite concentrations, that emissions of anthropogenic combustion Fe in global models could be significantly underestimated, and show that the atmospheric burden of Fe is potentially up to 8 times greater than previous estimates have suggested (Luo et al., 2008).

LFe can be also formed in the atmosphere during atmospheric processing of mineral dust and combustion aerosols (Ito, 2012; Ito and Feng, 2010; Johnson and Meskhidze, 2013; Meskhidze et al., 2005; Myriokefalitakis et al., 2015). We use here the general term "solubilisation" to describe the process that converts Fe from relatively insoluble minerals to the soluble Fe during atmospheric transport and photo-chemical transformation in the aqueous solution of aerosols and clouds.

Iron solubility (i.e., the fraction of total Fe that is soluble) in atmospheric aerosols over the Atlantic and Pacific Oceans has been observed to be in the range of 0.1–67% during oceanographic cruises (Baker et al., 2006a; Furutani et al., 2010), with even higher solubilities (up to 80%) measured in precipitation samples in the Southern Ocean (Heimburger et al., 2013). Acidic compounds (e.g., sulfates and nitrates) coating dust particles increase the Fe solubility during atmospheric transport of Fe-containing particles in model simulations (e.g., Meskhidze et al.,

2005). Indeed, since in direct measurements the fresh dust particles present low (<<1%) initial solubilities (Chuang et al., 2005; Fung et al., 2000; Hand et al., 2004; Sedwick et al., 2007), the atmospheric processing of dust (Kumar et al., 2010; Meskhidze et al., 2003; Srinivas et al., 2014) is considered as the best candidate to explain the high aerosol solubilities commonly observed at lower loadings (Baker and Jickells, 2006; Sholkovitz et al., 2012; Oakes et al., 2012). These processes may alter also the global pattern of LFe deposition (Fan et al.,

2004), especially within remote regions, such as the Atlantic, the Pacific (e.g., Sedwick et al., 2007) and the Southern Ocean (Ito and Kok, 2017; Johnson et al., 2010, 2011).

There is clear experimental evidence that atmospheric acidity - which is mainly driven by air pollution over high populated regions especially over the Northern Hemisphere (e.g., Seinfeld and Pandis, 2006) as well as natural sources such as volcanic sulfur emissions and the oceanic emissions of dimethylsulfide (DMS) in relatively

pristine ecosystems (e.g., Benitez-Nelson et al., 2003) - increases the dust solubility. Laboratory studies indicate that Fe solubilisation from minerals under acidic conditions in aerosol or rain droplets (Brandt et al., 2003; Shi et al., 2011; Spokes et al., 1994) occurs on different timescales; from hours to weeks depending on the size and the type of the Fe-containing minerals (Shi et al., 2011), with amorphous and ultrafine Fe solubilized much faster in acidic solutions (Brandt et al., 2003) compared to the aluminosilicates (roughly 10-14 days). Other laboratory

studies also support the occurrence of photo-induced reductive Fe solubilisation under acidic conditions (e.g., Fu et al., 2010), a mechanism that involves electron transfer to Fe(III) atoms on the particle surface to produce Fe(II)





(Larsen and Postma, 2001). The reductive solubilisation of minerals that are rich in Fe is also observed to be accelerated in the presence of Fe(II) or Fe(II)-ligand complexes (Litter et al., 1994). The oxalate-promoted solubilisation is controlled by the breaking of Fe-O bonds at the minerals surface due to the formation of a mononuclear bidentate ligand with surface Fe (Yoon et al., 2004), with the solubilisation rate significantly

increased as the pH decreases. Luo and Gao (2010) further prescribed pH and oxalate/hematite ratio dependent solubilisation rates for mineral dust, based on the laboratory experiments of Xu and Gao (2008). For weakly acidic conditions (pH=4.7) and various oxalate concentrations, a positive linear correlation between oxalate concentrations and the released LFe from different minerals has been also observed (Paris et al., 2011; Paris and Desboeufs, 2013). Laboratory investigations (Chen and Grassian, 2013; Siffert and Sulzberger, 1991) indicated

that even under highly acidic solutions (pH=2-3), the oxalic acid can be more important for the Fe solubilisation process of dust and combustion aerosols than the sulfuric acid through the formation of Fe(III)−oxalate complexes. Thus the minerals' solubilisation depends mainly on the proton concentration, the mineral surface concentration of organic ligands (such as oxalate), the sunlight and the ambient temperature (e.g., Hamer et al., 2003; Lanzl et al., 2012; Lasaga et al., 1994; Zhu et al., 1993).

The first modelling efforts that took into account the mixing of mineral dust with such anthropogenic acidic trace gases like sulfur dioxide ($SO_2$) (Fan et al., 2004; Meskhidze et al. 2003; Meskhidze et al., 2005; Solmon et al., 2009) showed considerable enhancements of atmospheric soluble Fe concentrations. A review by Mahowald et al. (2009) pointed out that human activity may have significantly modified the soluble Fe oceanic deposition flux, because anthropogenic combustion processes increased both Fe emissions and the acidity of atmospheric

aerosols. Recent studies (Meskhidze et al., 2017; Tagliabue et al., 2017) further show that atmospheric and oceanic organic ligands may increase the Fe solubilisation in the atmosphere and in the ocean, by forming Fe complexes that further increases the Fe bioavailability for the marine ecosystems. State-of-the-art global models clearly indicate a strong spatial and temporal variability of atmospheric LFe supply to the global ocean, that can be partly attributed to atmospheric processing. The global LFe deposition flux is currently estimated in the range

of 0.4–1.1 Tg-Fe $yr^{-1}$ (Ito and Kok, 2017; Ito and Shi, 2016; Ito and Xu, 2014; Johnson and Meskhidze, 2013; Luo et al., 2008; Myriokefalitakis et al., 2015; Wang et al., 2015).

In order to constrain a global picture of the influence of present atmospheric composition on the Fe supply to the oceans, we perform here a systematic comparison between models and between models and observations. We identify possible similarities and differences among models and between models and observations. The goals of

the present study are to (1) quantify the magnitude of the atmospheric TFe and LFe fluxes to the global ocean as calculated by four state-of-the-art global atmospheric aerosol models, (2) explain the differences of the simulated



LFe among the participating models, as well as (3) to provide multi-model ensemble TFe and LFe atmospheric Fe deposition fluxes for the next-generation of ocean biogeochemistry modelling studies.

The following discussion is organized in four sections: Section 2 describes the participating models and the observations used in this study. To build a concise view of the present-day understanding on the magnitude as well as the distribution of the TFe and LFe simulated deposition fluxes to the global ocean, ensemble model calculations are also presented. Section 3 presents and discusses the simulated global Fe atmospheric budgets and distributions. In Sect. 4, the uncertainties in the calculated surface aerosol Fe concentrations and deposition fluxes are discussed and the potential model biases are analyzed by attributing them to their major contributors. Finally, in Sect. 5 the findings of the present study are summarized together with recommendations for future research directions.

## 2 Methods

### 2.1 Description of models

The global models participating in this study differ in the spatial horizontal and vertical resolution, the meteorology, the emissions used for gas and aerosol species, as well as, the aerosol microphysics (i.e., size distribution and refractive properties). They also differ in the gas- and aqueous- phase chemical schemes and the parameterisations of atmospheric transport and deposition processes. The main characteristics of the participating models are summarized in Table 1. Note, however, that for this intercomparison no requirements of specific year, meteorological conditions or emission inventories have been set to the model simulations. Presented data are therefore mainly based on earlier published (or soon to be published) modeling experiments, which are evaluated and systematically analyzed here.

1. The Community Atmosphere Model version 4 (CAM4) is embedded within the National Center for Atmospheric Research (NCAR) Community Earth System Model version 1.0.5 (CESM 1.0.5; Hurrell et al., 2013). The CAM4 simulations are conducted with a horizontal resolution of 2.0° x 1.9° (longitude x latitude) and 56 vertical layers up to 2 hPa and forced by NASA's Goddard Earth Observing System (GEOS-5) meteorology. The emission data sets for anthropogenic activities, such as fossil fuel and biofuel combustion, are taken from the Aerosol Comparison between Observations and Models (AeroCom) database (Dentener et al., 2006). Desert dust is modeled following the Dust Entrainment and Deposition (DEAD) module (Zender et al., 2003) with updates in size fractions (Kok, 2011) and optics as described in (Albani et al., 2014). The bin widths are prescribed at 0.1-1.0, 1.0-2.5, 2.5-5.0, and 5.0-10.0



μm diameter and with fixed lognormal sub-bin distributions. Dust in CAM4 is speciated into six minerals, clays (illite, kaolinite and montmorillonite), feldspar, calcite and hematite (Scanza et al., 2015), with a total dust source of about 1767 Tg yr$^{-1}$ calculated for the present day. Further details on the CAM4 model used for this work are provided in Scanza et al. (2018) and references therein.

**2.** The GEOS-Chem model is driven by assimilated meteorological fields from the Goddard Earth Observing System (GEOS-5) of the NASA Global Modeling Assimilation at a horizontal 2.0°×2.5° (latitude x longitude) grid resolution and 47 vertical levels up to 0.01 hPa. GEOS-Chem simulates the emissions and chemical transformation of sulfur compounds, carbonaceous aerosols, and sea salt, and includes $H_2SO_4$-$HNO_3$-$NH_3$ aerosol thermodynamics solved by the ISORROPIA II thermodynamic

model (Fountoukis and Nenes, 2007) coupled to an $O_3$-$NO_x$-hydrocarbon-aerosol chemical mechanism. GEOS-Chem combines the DEAD scheme with the source function used in the Goddard Chemistry Aerosol Radiation and Transport (GOCART) model. Once mineral dust is mobilized from the surface, the model uses four standard dust bins with diameter boundaries of 0.2–2.0, 2.0–3.6, 3.6–6.0 and 6.0– 12.0 μm to simulate global dust transport and deposition, emitting 1614 Tg yr$^{-1}$ of mineral dust globally.

Further details on the GEOS-Chem model used for the this work can be found in Johnson and Meskhidze (2013) and references therein.

  **3.** The Integrated Massively Parallel Atmospheric Chemical Transport (IMPACT) model (Rotman et al., 2004) is also driven by assimilated meteorological fields from the Goddard Earth Observation System – Forward Processing (GEOS–FP) of the NASA Global Modeling and Assimilation Office (Lucchesi,

2017) with a horizontal resolution of 2.0°×2.5° and 59 vertical layers up to 0.01 hPa. The model simulates the emissions, chemistry, transport, and deposition of major aerosol species (Liu et al., 2005) and their precursor gases (Ito et al., 2007). IMPACT takes into account emissions of primary aerosols and precursor gases of secondary aerosols such as sulfate, nitrate, ammonium and oxalate. The emission data sets for anthropogenic activities such as fossil fuel use and biofuel combustion are taken from the

Community Emission Data System (CEDS) (Hoesly et al., 2018). Fe-containing combustion and dust aerosols are distributed among 4 bins in the model, with diameters: <1.26, 1.26–2.5, 2.5–5, and 5–20 μm, respectively (Ito, 2015; Ito and Feng, 2010). The present-day emission estimates for natural sources as well as combustion aerosols from biomass burning are used together with anthropogenic emissions (Dentener et al., 2006; Ito et al., 2018). A total dust source of 5070 Tg yr$^{-1}$ is dynamically calculated by a

physically-based dust emission scheme (Kok et al., 2014a, 2014b) in the model for the present day (Experiments 3 in Ito and Kok, 2017). The chemical composition of mineral dust and combustion



aerosols can change dynamically from that in the originally emitted aerosols due to reactions with gaseous species. Aerosol pH is calculated from the internal particle composition ($H^+$ and $H_2O$) for each size bin by the thermodynamic equilibrium module (Jacobson, 1999). The aerosol acidity depends on the aerosol types, mineralogy, particle size, meteorological conditions, and transport pathway of aerosols (Ito and Feng, 2010; Ito and Xu, 2014; Ito, 2015). A more detailed description of the IMPACT model used for this work can be found in (Ito, 2015; Ito and Kok, 2017; Ito and Shi, 2016) and references therein.

4. The TM4-ECPL global chemistry transport model simulates the oxidant ($O_3$/$NO_x$/$HO_x$/$CH_4$/CO) chemistry, accounting for non-methane volatile organic compounds, including isoprene, terpenes and aromatics, multiphase chemistry in clouds and aerosol water, as well as all major primary and secondary aerosol components, including sulfate, nitrate and secondary organic aerosols. TM4-ECPL is coupled with the ISORROPIA II thermodynamic model (Fountoukis and Nenes, 2007) and it uses modal size (lognormal) distributions to describe the evolution of fine and coarse aerosols in the atmosphere. Dust emissions, for the present version of the model, are calculated online based on the dust source parameterization of Tegen et al. (2002), as described in Myriokefalitakis et al. (2016); with the updated dust source calculations to produce slightly higher (~7%) dust emissions of around 1181 Tg yr$^{-1}$ compared to the modified AeroCom inventory (Dentener et al., 2006) taken into account in the previous version of the model (Myriokefalitakis et al., 2015). Dust is emitted in the fine and coarse mode with mass median radii (lognormal standard deviation) of 0.34 µm (1.59) and 1.75 µm (2.00), respectively. Note also that in the updated version of the model, the mineral-containing combustion aerosols are emitted with a number mode radius (lognormal standard deviation) of 0.04 µm (1.8) and 0.5 µm (2.0) for the fine and coarse mode, respectively (Dentener et al., 2006; Myriokefalitakis et al., 2016). All aerosol species in the model are subject to hygroscopic growth and removal processes that overall affect the mass median radius. The aerosol hygroscopic growth in the model is treated as a function of ambient relative humidity and the composition of soluble aerosol components and the uptake of water on aerosols change the particle size. TM4-ECPL model is driven by ECMWF (European Center for Medium-Range Weather Forecasts) Interim re-analysis project (ERA-Interim) meteorology and it has a horizontal resolution of 3.0° x 2.0° in longitude by latitude, 34 hybrid layers from the surface up to 0.1 hPa and a model time step of 30 min. Further details on the TM4-ECPL model used for this study can be found in Myriokefalitakis et al. (2015, 2016) and references therein.





### 2.1.1 Iron emission parameterizations

The primary Fe sources taken into account by the models can be roughly grouped as 1) mineral dust and 2) combustion sources. Various parameterizations or simplifications of Fe emission are adopted by the models, with the most important in the context of this paper being the Fe content and initial Fe solubility in emissions. The

mean Fe content in dust emissions, as well as the initial Fe solubility in emissions taken into account by the participating models, are presented in the supplement (Fig. S1 and Fig. S2, respectively). In more details:

1. **Mineral dust emissions:** Mineral-Fe primary sources are derived from the total mineral dust emissions, the fraction of specific Fe-containing minerals in dust emissions and the Fe content of each mineral (Table S1). CAM4 uses a soil mineralogy map and the Fe content in soils is estimated based on

mineralogical content (Claquin et al., 1999; Scanza et al., 2015, 2018; Zhang et al., 2015). GEOS-Chem and TM4-ECPL take into account the global soil mineralogy data set developed by Nickovic et al. (2012). GEOS-Chem prescribes an initial Fe solubility of 0.45% for the most reactive and poorly crystalline pool of Fe in desert top soils (Fig. S1), based on the synthesis of data from the Saharan and Sahel regions of northern Africa (Shi et al., 2012). The IMPACT model uses the mineralogy map and the

Fe content in soils as estimated by Journet et al. (2014). All the Fe-containing minerals in the model (i.e., hematite, goethite, illite, smectite, kaolinite, chlorite, vermiculite, and feldspars) are considered to be in the clay-sized (diameters <2 μm), with only goethite, chlorite, and feldspars to be also present in the silt-sized soils (diameters between 2-50 μm; Journet et al., 2014). The Fe content averaged in size bins 1–3 (3.6%) is higher than in the size bin 4 (2.3%). IMPACT applies an initial Fe solubility of 0.1% (Ito and

Shi, 2016) to the mineral dust aerosols emitted in the atmosphere (Fig. S1). In TM4-ECPL, the Fe content of the different Fe-containing minerals of dust (i.e., illite, kaolinite, smectite, goethite and hematite and feldspars) is based on the recommendations of Nickovic et al. (2013) and the assumption that is equally distributed between clay- and silt-sized soils, while for GEOS-Chem the Fe content of mineral dust was set to the widely accepted global mean value of 3.5% (Duce and Tindale, 1991). The

initial solubility of the emitted Fe-containing dust particles in TM4-ECPL is prescribed as 4.3% on kaolinite and 3% on feldspars emissions (Ito and Xu, 2014), while other minerals are considered to be emitted containing only insoluble Fe. The resulted annual global mean TFe content of emitted dust particles in TM4-ECPL is calculated to be 3.2% on average (Fig. S2).

2. **Combustion emissions:** All models but one (i.e., GEOS-Chem) include Fe-combustion emissions. These

are considered to be emitted from different combustion sectors with various initial Fe solubilities, with





the most important ones to be those from biomass burning, coal and oil combustion (Table S2). The CAM4 simulation includes the combustion Fe sources derived from industrial, biofuels (e.g., residential heating) and fires (sum of wildfires and anthropogenic biomass burning), as described in Luo et al. (2008), with an assumption that 4% is soluble at emission, and atmospheric processing occurs as for dust.

Shipping Fe emissions are not currently represented within CAM4. IMPACT takes into account Fe emissions from biomass burning, coal combustion and oil combustion (Ito et al., 2018), while an initial Fe solubility (58±22%) which is only applied to the primary Fe emission of ship oil combustion aerosols (Ito, 2015) assuming other Fe combustion emission sectors as insoluble. TM4-ECPL takes into account Fe emissions from biomass burning, coal combustion and oil combustion, based on the recommendations

of Luo et al. (2008) for biomass burning and coal combustion and by Ito (2013) for oil combustion, assuming fixed Fe-solubilities of 12% for biomass-burning Fe emissions, 8% for coal combustion and 81% for oil combustion from shipping. Note that none of the current models considered here take into account volcanic emissions, although it may be an important source of LFe to some regions of the ocean (e.g., Duggen et al., 2010).

**2.1.2 Iron solubilisation parameterizations**

The conversion of insoluble-to-soluble Fe in the models can be parameterized as an aqueous-phase kinetic process that depends on (1) the proton activity (also termed as acid-promoted solubilisation), (2) the oxalate concentration (also termed as oxalate-promoted Fe solubilisation) and (3) the actinic flux (also termed as photo-reductive solubilisation). The simplification of the applied parameterisations differs among models; from the

models used in this study, only IMPACT takes into account all three solubilisation processes, but only in aerosol water for both dust and combustion aerosols. TM4-ECPL and GEOS-Chem apply an acid- and oxalate-solubilisation scheme only for dust aerosols, both in aerosol and cloud water.

CAM4 accounts for the atmospheric processing of both dust and combustion aerosols, based on the acid-(Meskhidze et al., 2005) and oxalate- (Paris et al., 2011) driven solubilisation processes in a simplified manner

appropriate for use in an earth system model (Scanza et al., 2018). The method is described in more detail in Scanza et al. (2018), but generally the acid promoted iron dissolution depends explicitly on modeled temperature and an assumed acidity, which is either high (i.e., pH=2) or low (i.e., pH=7.5) based on the relative model concentrations of sulfate and calcite, while the oxalate concentrations in cloud water required for ligand promoted iron dissolution are not explicitly calculated, but instead assumed to be proportional to the modeled

organic carbon aerosol concentration. CAM4 also assumes Fe from dust to be in either a slow, medium or readily





soluble state based on Shi et al. (2011) and Ito and Xu (2014), while Fe from combustion is assumed to be in a medium soluble state.

The other three models (i.e., GEOS-Chem, IMPACT and TM4-ECPL) calculate the proton-promoted solubilisation rate of minerals by applying an empirical parameterization of Meskhidze et al. (2005) and Johnson

and Meskhidze (2013), that takes into account the degree of saturation of the solution, the type of each mineral and the ambient temperature. The thermodynamic equilibrium modules are used to estimate the water content in the aqueous phase of hygroscopic particles (Jacobson, 1999; Fountoukis and Nenes, 2007). In addition to the mineral types, IMPACT and TM4-ECPL consider three dust-Fe pools associated with mineral source materials as measured by Ito and Shi (2016) and Shi et al. (2011), respectively, and the solubilisation rates calculated by Ito

and Shi (2016) and Ito and Xu (2014), respectively. Despite the different mineral databases used by the two models (see Sect. 2.1), the three Fe-pools are roughly similarly characterized in the models as ferrihydrite, nano-sized Fe oxides and heterogeneous inclusion of nano-Fe grains in aluminosilicates, respectively. For GEOS-Chem, the Fe containing mineral (i.e., hematite, goethite, and illite) solubilisation rate is based on the temperature-dependent equations of Meskhidze et al. (2005) and Johnson and Meskhidze (2013).

For the oxalate-promoted solubilisation, CAM4, GEOS-Chem and TM4-ECPL apply a linear relationship between solubilisation rates and oxalate concentration in the solution, based on the laboratory data of Paris et al. (2011), who measured the initial soluble Fe release rates of Fe-oxides and aluminosilicates (i.e., at pH = 4.7, and for one hour). TM4-ECPL applied this oxalate-solubilisation relationship for three Fe-containing minerals (hematite, goethite, and illite), using illite as a proxy for all Fe-containing aluminosilicate minerals (Johnson and

Meskhidze, 2013). In TM4-ECPL the formation of oxalate in cloud and aerosol water is explicitly simulated in the model (Myriokefalitakis et al., 2011), in contrast to GEOS-Chem in which the sulfate concentrations are used as a proxy for the oxalate production (Yu et al., 2005). However, in TM4-ECPL the oxalate Fe-solubilisation is applied only in cloud droplets, where in GEOS-Chem it is applied both in cloud and aerosol water. IMPACT also takes into account an explicit scheme of oxalate formation both in cloud and aerosol water (Lin et al., 2014),

applying however the oxalate-promoted Fe-solubilisation only in aerosol water (Ito, 2015). The constants used to calculate these Fe solubilisation rates in IMPACT are fitted to experimental data for coal fly ash (Chen and Grassian, 2013), while the rate of the photo-induced solubilisation is based on the Fe-dissolution rates of coal fly ash (Chen and Grassian, 2013), scaled on the photolysis rate of $H_2O_2$ estimated in the model.



### 2.1.3 Deposition parameterizations

The dry and wet deposition are considered as loss processes for all Fe-containing aerosols in the models. For this work, the dry deposition fluxes include both the gravitational settling and the turbulent deposition and the wet deposition takes into account both the in-cloud nucleation scavenging and the below- cloud scavenging in all

models. For CAM4, the dry removal of dust aerosols involves parameterizations for gravitational settling and turbulent mix out, and wet removal includes in-cloud and below-cloud scavenging (Rasch et al., 2000; Zender et al., 2003). For GEOS-Chem, the removal of mineral dust occurs through dry deposition processes such as gravitational settling (Seinfeld and Pandis, 2006) and turbulent dry transfer of particles to the surface (Zhang et al., 2001). Dust removal by wet deposition processes includes both convective updraft scavenging and

rainout/washout from large-scale precipitation (Liu et al., 2001).

For IMPACT, the dry deposition of aerosol particles uses a resistance-in-series parameterization (Zhang et al., 2001). Gravitational settling is also taken into account (Rotman et al., 2004; Seinfeld and Pandis, 2006). Aerosols and soluble gases can be incorporated into cloud drops and ice crystals within cloud (rainout), collected by falling rain and snow (washout), and be entrained into wet convective updrafts (Ito et al., 2007; Ito and Kok, 2017; Liu

et al., 2001; Rotman et al., 2004). The aging of dust and combustion aerosols from hydrophobic to hydrophilic enhances their dry and wet deposition. Hygroscopic growth of mineral dust and combustion aerosols in gravitational settling uses the (Gerber, 1991) scheme, including the particle growth due to sulfate, ammonium and nitrate associated with the particles (Liu et al., 2005; Xu and Penner, 2012). Scavenging efficiencies for mineral dust and combustion aerosols in wet deposition are calculated based on the amount of sulfate, ammonium

and nitrate coated on the particles (Liu et al., 2005; Xu and Penner, 2012). For TM4-ECPL, the dry deposition parameterizations are based on an online scheme that takes into account series of surface and atmospheric resistances (Ganzeveld and Lelieveld, 1995). The aerosol hygroscopic growth in the model is treated as a function of ambient relative humidity and the composition of soluble aerosol components (Gerber, 1985) changing thus the particle size and impact on aerosols gravitational settling. For the wet deposition in TM4-

ECPL, both the liquid and ice precipitation are taken into account, with a distinction between scavenging due to large-scale and convective precipitation. In-cloud scavenging in stratiform precipitation uses an altitude dependent precipitation formation rate and the scavenging efficiency is calculated taking into account the aerosols lognormal distributions. Note that in TM4-ECPL, all soluble aerosols are assumed to be completely scavenged in the convective updrafts producing rainfall rates of >1 mm/h, and exponentially scaled down for

lower rainfall rates.





### 2.2 The ensemble model

Ensemble model calculations of this study aim overall to provide robust results of the simulated atmospheric Fe concentrations and deposition fluxes. For these calculations, all fields for TFe and LFe in mineral dust and combustion aerosols (as well as for dust aerosols) are first converted to a common $1.0^{\circ} \times 1.0^{\circ}$ horizontal
resolution grid, by using the freely available Climate Data Operators (CDO v.1.9.1) software. The CDO is a collection of operators for standard processing of climate and forecast model data developed by the Max Planck Institute for Meteorology and for this work were applied with a bilinear interpolation to all fields, assuring an exact mass conservation. Further details about CDO can be found online in https://code.mpimet.mpg.de/projects/cdo/embedded/cdo.pdf.

The ensemble atmospheric concentrations and deposition fluxes of mineral dust Fe have been calculated from 4 models, while for the Fe originating from combustion sources, 3 models were used (see Table 2). For Fe-contained combustion aerosols, we simply use the mean of the respective fields of each model to derive the ensemble model, since no considerable differences appeared among the participating models. However, since model simulations of the global dust cycle are well known to have substantial biases in the size distribution
relative to *in situ* measurements and remote sensing observations (e.g., Huneeus et al., 2011; Kok et al., 2017; Ridley et al., 2016), we here attempt to reduce these biases by correcting the loading and deposition flux in each model's particle bin, using state-of-the-art constraints on size-resolved dust loading, as recently derived in Kok et al. (2017).

Specifically, a correction factor $c_{i,j}$ is applied to each particle bin $j$ of model $i$, which equals:

$$c_{i,j} = \frac{\int_{D_{i,j-}}^{D_{i,j+}} \frac{dM_{atm}}{dD} dD}{L_{i,j}} \tag{1}$$

where, $\frac{dM_{atm}}{dD}$ is the mass size distribution of the global atmospheric dust (see Figs. 2b and S1b in Kok et al. (2017)). This mass size distribution was obtained from measurements, modeling, and remote sensing constraints on the size distribution of emitted dust, the atmospheric lifetime and extinction efficiency of atmospheric dust, and the global dust aerosol optical depth (Kok et al., 2017; Ridley et al., 2016). Furthermore, $D_{i,j-}$ and $D_{i,j+}$ are
respectively the lower and upper size limits of particle bin $j$ in model $i$, and $L_{i,j}$ is the (not bias corrected) simulated global dust loading in that particle bin. Since emission and deposition fluxes scale with atmospheric dust loading, we correct these fluxes as well as the atmospheric load in each particle bin of each of the contributing models by multiplying the flux by the correction factor of Eq. (1). Note, however, that the bias correction calculated in this work is expected to correct only the part of the regional bias that stems from a bias in



the global deposition fluxes. Biases in the regional scales are affected by biases on the global scale, but also by other biases, for instance caused by uncertainties in the deposition scheme. The global mean bias correction factors (i.e., median, lower 95% confidence interval and upper 95% confidence interval) for each model and each aerosol size (bin or mode) are presented in Table 3.

For TM4-ECPL, which uses modal size distributions for dust (see Table 2), we also redistributed the fine and coarse aerosols into 4 bins, in order to apply the same methodology. Specifically, the new aerosol modes are re-calculated using the error function and based on the characteristic (radius and sigma lognormal) of each mode (see Table 2) to derive binned data for diameters bin 1: 0-1 μm, bin 2: 1-4 μm, bin 3: 4-10 μm and bin 4: 10-20 μm. This bias correction indicates biases mainly in the small modes, with the median correction factors for bins 1

- 4 being 0.134, 0.692, 1.257, and 1.81, respectively. Note also that for the ensemble model calculations of this study, the TFe and LFe depositions fluxes have been calculated as the sum of Fe from the corrected mineral dust and the mean combustion aerosols.

## 2.3 Iron atmospheric observations

To evaluate the models' ability to reproduce the observed distributions of surface TFe and LFe aerosol

concentrations over oceans, the model results have been compared with available observations from Achterberg et al. (2018), Baker et al. (2006a, 2006b, 2007, 2013), Baker and Jickells (2017), Bowie et al. (2009), Buck et al. (2006, 2010, 2013, pers. com. 2018), Chance et al. (2015), Gao et al. (2013), Guieu et al. (2005, pers. com. 2018), Jickells et al. (2016), Kumar et al. (2010), Longo et al. (2016), Powell et al. (2015), Shelley et al. (2015), Shelley et al. (2018), Sholkovitz et al. (2012), Srinivas et al. (2012), Srinivas and Sarin (2013), Wagener (2008)

and Wagener et al. (2008). A total of 818 observations of TFe and 795 daily observations of LFe over the ocean performed from September 1999 to March 2015 have been used for this purpose. The global distribution of the observed aerosol Fe-concentrations used for this study is presented in Fig. S3 and the respective coordinates in Table S3. A bulk dust deposition flux data set compiled by Albani et al. (2014) is also used for comparison of Fe deposition flux after multiplying by the averaged Fe content in upper crustal minerals of 3.5%.

TFe were obtained from sampling aerosols in air mainly on board oceanographic cruises, and for some studies at sampling stations located on shore or on islands with no local anthropogenic influence. A variety of samplers (small or high volume) have been used to collect particulate Fe, using different types of filters. TFe was either measured on whole filter or part of filter by X-ray fluorescence or after acid digestion. LFe was obtained following several protocols, e.g. contact time, volume of media, and type of media (ultra-pure water or filtered

surface seawater for different pH conditions and various amounts of Fe ligands). As highlighted in Baker and



Croot (2010), these diverse experimental approaches used for determination of aerosol Fe solubility cause part of the variability observed. Although there is still no consensus in the experimental way to determine LFe, this data set provides valuable and robust data in all oceanic region to be compared to model outputs. Here the results from the different protocols are all combined together. Overall, the results are analyzed with regard to the role of the different model complexities, providing insight to directions for future model improvements.

## 3 Results

### 3.1 Global budgets

All participating models have submitted results to enable the analysis of the TFe and LFe global budgets and atmospheric concentrations, both for dust and combustion aerosols, for emissions, dry and wet deposition fluxes, atmospheric processing and atmospheric loads (see Sect. 2). Concerning the temporal resolution, daily mean spatially-resolved budget terms have been submitted for IMPACT, while for GEOS-Chem, CAM4 and TM4-ECPL monthly mean fields for all budget terms are provided. For atmospheric concentrations however, all models provided daily mean fields. Concerning the aerosol size distribution (see Table 2), IMPACT and CAM4 submitted budget fields for four size bins, TM4-ECPL for two size modes, while GEOS-Chem provided results as bulk aerosols. CAM4 and TM4-ECPL submitted separate fields for proton and oxalate Fe solubilisation, while IMPACT and GEOS-Chem provided total fields. IMPACT and CAM4 have also submitted atmospheric processing terms for dust and combustion aerosols, while TM4-ECPL only for dust aerosols since no solubilisation processes are calculated for Fe combustion aerosol in the model. GEOS-Chem does not take into account Fe from combustion aerosols.

### 3.1.1 Iron sources and deposition

The computed TFe and LFe emissions together with the deposition fluxes for all models are presented in Table 4. Note however that for LFe, both from mineral dust and combustion sources, the total sources (sum of primary and secondary sources) are here discussed rather than the primary emissions alone. The models use significantly different assumptions to describe the total LFe source to the atmosphere and therefore primary (emissions) and secondary (atmospheric processing) sources cannot be accurately separated from rapid formation assumed in coarse-scale models. The computed annual TFe emissions and LFe sources (emissions and atmospheric processing) from 1) mineral dust and 2) combustion sources for each model are presented in Table 4 and the corresponding global emission/sources distributions are shown in Fig. S4 and Fig. S5, respectively.



The importance of wet versus dry deposition as removal processes for atmospheric aerosols depends on the aerosol solubility and size distribution (and the presence/amount of precipitation). Focusing however on the deposition to oceans for this study, the apportionment of the total atmospheric deposition within different oceanic regions is presented in Table 5 (for this deposition analysis we use here the ocean classification as provided by HTAP phase-2: available online via the HTAP Wiki). Moreover, the computed annual deposition flux distributions of TFe and LFe from mineral dust and combustion sources together for each model are further presented in Fig. S6.

Overall, the Fe sources and deposition in the models are here classified as:

1. **Total Fe:** The modelled annual mean emission fluxes of TFe from mineral dust (TFeD) are calculated to be in the range of 38 - 134 Tg-Fe yr$^{-1}$. CAM4 and GEOS-Chem calculate similar annual TFeD emission fluxes (around 57 Tg-Fe yr$^{-1}$), TM4-ECPL emissions are about 40% lower (~38 Tg-Fe yr$^{-1}$) and IMPACT is about 2.4 times higher (around 134 Tg-Fe yr$^{-1}$). Note that IMPACT takes into account the largest flux among the participating models (Table S1) mainly due to the largest upper size. However, dust fluxes in the same size range for the different models are comparable after the bias correction based on the analysis by Kok et al. (2017) (see Sect. 2.2). TFe emissions from combustion sources (TFeC) range between 1.8 and 2.7 Tg-Fe yr$^{-1}$, with CAM4 and TM4-ECPL calculating annual mean fluxes of around 1.8 Tg-Fe yr$^{-1}$ globally, and IMPACT having a 35% higher estimate. On a global scale, dry deposition is the most important removal mechanism, across all models for the TFeD; IMPACT has the highest dry deposition flux of all models (~68 Tg-Fe yr$^{-1}$), followed by GEOS-Chem (~40 Tg-Fe yr$^{-1}$), CAM4 (~33 Tg-Fe yr$^{-1}$) and TM4-ECPL (~30 Tg-Fe yr$^{-1}$). Submicron aerosols are removed mostly by wet removal while for supermicron aerosols the gravitational settling is important (e.g., Seinfeld and Pandis, 2006). Consequently, the wet removal of TFeD across almost all models (except for IMPACT) is smaller than dry deposition flux - mainly due to the high contribution of coarse aerosol sedimentation to the dry removal processes (Table 4). The simulated wet deposition flux of TFeD ranges over about one order of magnitude (from about 8 to 66 Tg-Fe yr$^{-1}$); IMPACT calculates the highest TFeD wet deposition flux of all models of about 66 Tg-Fe yr$^{-1}$ (~49% of total removal), followed by CAM4 (24 Tg-Fe yr$^{-1}$; 42%), GEOS-Chem (16 Tg-Fe yr$^{-1}$; ~29%) and TM4-ECPL of roughly 8 Tg-Fe yr$^{-1}$ (~20%). In contrast to TFeD, due to the similar assumptions of the size distribution and scavenging efficiency in the models, the wet deposition is the larger removal pathway for TFe from combustion processes (TFeC), (except for TM4-ECPL, probably due to the different solubility factors in primary emissions and the different



atmospheric processing parameterizations), responsible overall for about 60% of the total TFeC removal across models and amounting to about 1 Tg-Fe yr$^{-1}$.

2. **Labile Fe:** The global annual mean LFe sources from mineral dust (LFeD) range between 0.3 and 1.0 Tg-Fe yr$^{-1}$. IMPACT and GEOS-Chem calculate similar LFeD sources, close to 0.7-0.8 Tg-Fe yr$^{-1}$, where CAM4 calculates the highest annual source and TM4-ECPL the lowest. However, these differences are mainly attributed to the secondary processes leading to LFeD production rather than the primary emissions. For example, despite the large difference between IMPACT and TM4-ECPL in TFeD sources, the models consider similar emission amounts of LFeD emissions of about 0.12-0.13 Tg-Fe yr$^{-1}$. In contrast, the secondary LFeD produced due to atmospheric processing is calculated to vary by a factor of 3-4 between 0.57 Tg-Fe yr$^{-1}$ (IMPACT) and 0.17 Tg-Fe yr$^{-1}$ (TM4-ECPL), respectively. CAM4 takes into account LFeD emissions of around 0.18 Tg-Fe yr$^{-1}$, but the highest annual LFeD atmospheric processing (0.8 Tg-Fe yr$^{-1}$), and GEOS-Chem about 0.25 Tg-Fe yr$^{-1}$ and 0.54 Tg-Fe yr$^{-1}$ for LFeD emissions and atmospheric processing, respectively. The LFe source from combustion aerosols (LFeC), with a range of about 0.1-0.2 Tg-Fe yr$^{-1}$, shows smaller differences than that from mineral dust (0.3-1.0 Tg-Fe yr$^{-1}$). Although the differences are not large, these clearly depict the different assumptions followed by these two models; IMPACT does not account for primary LFe sources from combustion (expect those from oil ship combustion of about 0.009 Tg-Fe yr$^{-1}$), thus almost all the LFeC sources over land are attributed to secondary production via atmospheric processing (0.091 Tg-Fe yr$^{-1}$). In contrast, TM4-ECPL does not take into account atmospheric processing of Fe from combustion sources, attributing all the LFeC sources to direct emissions (~0.2 Tg-Fe yr$^{-1}$). Finally, CAM4 that includes both direct emissions and atmospheric processing of LFeC, calculates a total source of about 0.13 Tg-Fe yr$^{-1}$, corresponding to roughly 0.075 Tg-Fe yr$^{-1}$ and 0.053 Tg-Fe yr$^{-1}$, for primary LFeC emissions and atmospheric processing respectively.

### 3.1.2 Iron seasonal variability

Figure 1 presents the global LFe sources (positive) and oceanic deposition fluxes (negative) for all participating models and their ensemble mean (see Sect. 3.2), for the four seasons, i.e., December, January and February (DJF), March, April and May (MAM), June, July and August (JJA) and September, October and November (SON). LFe sources are mainly driven by mineral dust aerosols, although a significant fraction (6 to 62%) is due to LFe combustion aerosols, especially over the high-latitudes of the Northern Hemisphere (Ito et al. 2018; companion manuscript to be submitted). For LFe sources, despite the different assumptions applied in the models





(i.e., atmospheric processing and direct LFe emissions), maximum sources are calculated for MAM and JJA due to intense dust emissions and biomass burning, respectively. The models with the highest LFe sources, also exhibit the highest deposition fluxes to the ocean. However, significant differences in the magnitude of the deposition fluxes are calculated between models, mainly for DJF and JJA. A seasonal maximum in the deposition fluxes is calculated by CAM4 during MAM, attributed to Saharan mineral dust aerosols, and in most of the cases IMPACT and GEOS-Chem present similar seasonal variation.

Figure S7 (supplement) further presents the zonal mean seasonal variability of the LFe global sources and oceanic deposition fluxes. Most of LFe emissions are calculated to occur over the mid-latitudes of the Northern Hemisphere (NH) for all seasons, with maximum during MAM and JJA and minima during SON. In DJF two zonal maxima are shown near the equator and around 30N, while in the other seasons the 30N maximum is not clearly present. The equatorial maximum in DJF is shifted to the NH in JJA following the Intertropical Convergence Zone (ITCZ) migration and the subsequent geographic change in the location of biomass burning emissions. Again, all models appear to have similar LFe seasonality, with the highest LFe oceanic deposition fluxes across all models calculated by CAM4 (Table 5).

## 3.2 Ensemble model calculations

### 3.2.1 Iron surface concentrations

The annual mean surface TFe aerosol concentrations for the ensemble model exceed 100 μg-Fe m$^{-3}$ over the major dust source regions such as the Sahara Desert, where mineral dust particles dominate the atmospheric Fe burden (Fig. 2a). Relatively high TFe concentrations (e.g., up to 10 μg-Fe m$^{-3}$ over the tropical Atlantic Ocean) are calculated for ocean regions at the outflow from these source regions. High TFe concentrations of around 6 μg-Fe m$^{-3}$ are also calculated over heavily polluted areas like China, while secondary maxima up to 2-5 μg-Fe m$^{-3}$ are calculated over the central Africa, Asia and Indonesia, where Fe-containing aerosols are associated with biomass burning emissions (Fig. 2b).

Model Fe solubility calculations (Fig. 2c) clearly suggest the impact of atmospheric processing on the derived LFe ensemble surface concentrations, with high Fe solubilities calculated far from source regions over the remote tropical oceans, corresponding to low TFe concentrations. Ensemble annual mean LFe concentrations around 0.5 μg-Fe m$^{-3}$ occur downwind of the Sahara and around 0.1 μg-Fe m$^{-3}$ downwind of the Arabian and Gobi Deserts. At the outflow of these regions, the Fe solubility over the global ocean is calculated to be about 1-1.5%, with the highest Fe solubilities (4-5%) over the tropical Atlantic Ocean (Fig. 2c). Additionally, LFe concentrations over





polluted regions may range up to 0.05 µg-Fe m$^{-3}$, indicating a significant anthropogenic contribution via direct combustion emissions and atmospheric processing (Fig. 2b). Over central South America, Asia and Indonesia, LFe concentrations of about 0.03-0.05 µg-Fe m$^{-3}$ (corresponding to high Fe solubilities up to 5%) are found due to both direct biomass-burning emissions and due to ligand-promoted dissolution. The latter process is enhanced

in these areas by the respective enhanced oxalate production upon the oxidation of emitted biogenic VOCs precursors, such as isoprene, under cloudy conditions (Lin et al., 2014; Myriokefalitakis et al., 2011).

### 3.2.2 Iron deposition fluxes

Model calculations indicate that about 71.5 (± 43) Tg-Fe yr$^{-1}$ of TFe from mineral dust are deposited to the Earth's surface (Table 4), with ensemble deposition fluxes of around 5000-8000 mg-Fe m$^{-2}$ yr$^{-1}$ calculated

downwind of the main desert source regions (Fig. 3a). However, within the Northern Atlantic Ocean in the outflow of the Sahara, the model mean indicates deposition fluxes up to 2400 mg-Fe m$^{-2}$ yr$^{-1}$, while within the Northern Pacific Ocean in the outflow of Gobi Desert and within the Southern Ocean downwind of the Patagonia Desert the ensemble model shows annual mean fluxes of ~34 and ~10 mg-Fe m$^{-2}$ yr$^{-1}$, respectively (Fig. 3a). The TFe annual mean global deposition flux from combustion sources (Table 4) is calculated to be about 2.2 (± 0.5)

Tg-Fe yr$^{-1}$, with two main regions where TFe concentrations exceed 2500 mg-Fe m$^{-2}$ yr$^{-1}$, one near biomass burning regions (e.g., southern Africa, South America and southeast Asia) with up to ~3000 mg-Fe m$^{-2}$ yr$^{-1}$, and a second near highly populated regions with Fe released from coal and oil combustion processes (India and China) with up to ~3500 mg-Fe m$^{-2}$ yr$^{-1}$ (Fig. 3b).

A global mean LFe deposition flux of 0.7 (± 0.2) Tg-Fe yr$^{-1}$ is derived from all models (Table 4), with about one

third (~0.24 Tg-Fe yr$^{-1}$) calculated to be deposited to the global ocean for the ensemble model (Table 5). The highest annual mean LFe deposition fluxes (up to 36 mg-Fe m$^{-2}$ yr$^{-1}$) are simulated within dust source regions (Fig. 3c), owing mainly to the LFe content of the emissions (e.g., see Fig. 2d). The global model-mean LFe deposition fluxes from combustion sources are calculated at about 0.2 (± 0.04) Tg-Fe yr$^{-1}$ (Table 4), with maximum global deposition rates of 4-5 mg-Fe m$^{-2}$ yr$^{-1}$ (Fig. 3c) simulated in the outflow of tropical biomass

burning regions (i.e., South America, Africa and Indonesia), clearly reflecting the contribution of combustion processes. Focusing on the marine environment, annual mean LFe deposition rates of 15 mg-Fe m$^{-2}$ yr$^{-1}$ are calculated for the tropical Atlantic Ocean and for the Indian Ocean (up to 16 mg-Fe m$^{-2}$ yr$^{-1}$) under the influence of the Arabian and Indian peninsulas, but up to ~29 mg-Fe m$^{-2}$ yr$^{-1}$ for the Mediterranean Sea downwind of the Sahara Desert. Deposition rates around 1 mg-Fe m$^{-2}$ yr$^{-1}$ are calculated to occur within the Northern Pacific in the

outflow from the Gobi Desert as well as within the Southern Hemisphere downwind of Patagonia to the Southern





Ocean (up to 0.1 mg-Fe m$^{-2}$ yr$^{-1}$) and downwind of the dust source regions of Australia and South America (up to ~4 mg-Fe m$^{-2}$ yr$^{-1}$). The LFe deposition rates to the Southern Ocean are associated mainly with the Patagonian, Southern African and Australian deserts, with a smaller contribution in the subtropical ocean from biomass burning sources (Fig. 3d). Note, that the largest fluxes of LFe deposited to the HNLC region of the

Southern Ocean (e.g., south of the Antarctic circumpolar current) are simulated to be originating from a Patagonian mineral dust source, with rates reaching 0.1 mg-Fe m$^{-2}$ yr$^{-1}$. The ensemble annual mean deposition fluxes to various oceanic regions are further presented in Table 5.

## 4 Uncertainties

### 4.1. Comparison with measurements

The TFe loading, Fe solubility, and LFe loading from the models are compared with the measurements and presented in Fig. 4. All models captured a tendency of higher Fe loading near and downwind of the major dust source regions. The mean normalized biases (MNB) between the models and observations are also presented. The MNB for the Northern Hemisphere (14 for ensemble model) is larger than that for the Southern Hemisphere (2.4 for ensemble model). This reflects that most models overestimate TFe surface mass concentrations near the dust

source regions and tend to underestimate the concentrations observed over remote oceans. This might also indicate an inaccurate transport from the continental source areas to the observational sites or a too fast loss via deposition in the models. Overall, bias correction for ensemble model improves agreement of the ensemble model against measurements (Fig. S8). We note, however, that matching the atmospheric concentrations may cause a high bias in simulated Fe depositions at low values in the SH (Albani et al., 2014; Huneeus et al., 2011)

(Fig. S9). The computed correlation coefficients of the ensemble model against measurements at the surface are 0.13 for TFe, 0.05 for Fe solubility, and 0.25 for LFe, respectively, which are much smaller than those between the participating models (0.57–0.90 for TFe (GEOS-Chem vs. IMPACT)–(CAM4 vs. TM4-ECPL), 0.05–0.56 for Fe solubility (CAM4 vs. TM4-ECPL)–(CAM4 vs. IMPACT), and 0.40–0.75 for LFe (GEOS-Chem vs. IMPACT)–(CAM4 vs. GEOS-Chem)). This indicates a linear dependence of the model results and that the

models have similar behavior accounting for the same key processes that affect Fe deposition. The small positive correlation between models and observations indicates that the models miss –or do not accurately represent-important processes that drive the variability in the observations. Indeed, when comparing the computed and observed solubilities of Fe, all models overestimate the lowest solubilities (<0.1%) observed close to the source regions (14 samples in Arabian Sea and 2 samples in tropical Atlantic). This is primarily due to the assumed



solubility of the dust aerosols at emissions, and the subsequent enhancement of Fe solubility estimated from the simulated amount of atmospheric processing during transport in the models. It is noted, however, that the lowest solubilities in the measurements are outliers in the negative slope of Fe loading versus Fe solubility in the NH (Figure 5).

All models underestimate the high end of the observed values (>10%) in the Southern Ocean, which are mainly associated with transported and aged aerosols and this is potentially a significant shortcoming because this is an oceanic HNLC region where atmospheric Fe supply has a potentially important impact on ocean productivity in past and future climate. In GEOS-Chem, IMPACT, and TM4-ECPL, Fe dissolution over the Southern Ocean is suppressed mainly due to the lack of anthropogenic emissions and the subsequent acidification of the aerosols.

CAM4 is relatively insensitive to the acidity, since most labile Fe is formed by in-cloud processes. Thus, the model results from CAM4 can in part test whether in-cloud processing can realistically describe the observed pattern of solubility over the Southern Ocean. CAM4 shows higher Fe solubility than field data for most samples in the Southern Ocean (69% in CAM4, 7% in GEOS-Chem, 55% in IMPACT and 5% in TM4 TM4-ECPL) (Figs. 4 and 5). Thus, the wide range in observed Fe solubility cannot be explained by excluding the effect of modelled

aerosol acidity over the Southern Ocean. It is worth mentioning that IMPACT, which has the highest complexity in simulating labile Fe, reproduces the widest range of observed Fe solubility. It should be noted, however, that the comparison of monthly mean model results with the shorter-term (e.g., daily) observations during different sampling periods introduces inaccuracies due to an episodic nature of high Fe solubility. A more detailed comparison of Fe solubility between models and observations is presented in a separate companion paper to this

work (Ito et al. 2018; in preparation).

## 4.2. Model-to-model comparison

Model budget analysis and model evaluation indicate that even though the models are able to reproduce surface Fe measurements to some extent, large differences existed among modes in processes such as emissions, transport and deposition. A large diversity is here documented between models in terms of LFe primary sources

(i.e., emissions) and secondary processes (i.e., atmospheric processing) which introduce uncertainties in the estimated oceanic deposition. There are many intrinsic reasons, however, for this diversity in the Fe simulations among models; besides the emitted Fe mass in the atmosphere (especially from dust aerosols), the aerosol size distribution, the soil mineralogy, the strength of combustion aerosol sources, as well as the parameterizations used to calculate the pH of the aerosol water and the oxalate production are large sources of uncertainty in model

simulations.



The aerosol size and solubility are important factors driving the atmospheric cycle of Fe, since they both control the removal processes from the atmosphere via the dry deposition (including gravitational settling) and the wet scavenging (e.g., Albani et al., 2014). It is well documented that the lifetime of Fe-containing aerosols ranges between less than one day for the coarse mode (with diameters larger than 1 µm) particles to weeks for the fine

mode (with diameters less than 1 µm) (e.g., Ginoux et al., 2012; Luo et al., 2008; Mahowald et al., 2009; Tegen and Fung, 1994), with the overall lifetime of dust aerosols to usually range between 1.6 and 7.1 days in the models (Huneeus et al., 2011). Fine particles have longer lifetimes and thus experience more atmospheric processing. The conversion of insoluble minerals content to soluble forms as a result of aging during atmospheric transport increases aerosols' solubility. Lifetime calculations can thus provide a valuable tool to determine the Fe

persistence in the atmosphere, overall integrating sources, transport and deposition differences among models, especially over remote areas such as the open ocean.

### 4.2.1 Iron atmospheric lifetime (turnover time)

Figure 6 presents the spatial distribution of TFe lifetime over the ocean as calculated for the ensemble model. For TFe originating from dust sources, the calculated global mean lifetime over the oceans is ~6 (± 4) days (Fig. 6a).

The lifetime of combustion TFe over oceans is longer, at around 14 (± 9) days (Fig. 6c), due to the overall smaller size of the combustion Fe aerosols compared to that of mineral dust (affecting their sedimentation processes, their horizontal and vertical transport in the atmosphere), but also because of the low precipitation over part of the regions in which mineral dust and combustion aerosols are transported. The ensemble model indicates long TFe lifetimes over remote oceanic regions, such as in the outflow of South America, in the outflow

of South Africa, as well as in the outflow of Australia (Fig. 6c). Over these so-called ocean deserts, where precipitation is low and thus the wet deposition rates are low, the ensemble model results overall in longer lifetimes for Fe containing aerosols, although the LFe atmospheric concentrations can be extremely low.

To further analyze the differences among the models, the standard deviation (STD) of the TFe lifetime is calculated for each grid for dust aerosols (Fig. 6b), combustion aerosols (Fig. 6d), and the combined TFe lifetime

(Fig. 6f). Over remote oceanic regions, the high STD can be related to the different assumptions used by the models to parameterize 1) the long-range transport of Fe-containing aerosols of different parameterizations of the sizes, 2) the wet and dry deposition parameterizations, and 3) the soluble fraction of Fe-containing combustion aerosols (e.g., the differences in ship oil combustion and biomass burning emissions). From the models that include Fe from combustion processes, IMPACT assumes that only ship oil combustion emissions have an initial

fraction of soluble Fe, and thus all the LFe from continental sources is produced due to atmospheric processing.





CAM4 does not take into account ship emissions but includes both primary and secondary sources for LFe from other combustion sources. TM4-ECPL although includes continental and ship oil combustion emissions both for TFe and LFe but, it does not take into account any dissolution processes for combustion aerosols, although continental and ship oil combustion emissions both for TFe and LFe are considered. Differences in the

precipitation patterns or parameterization for wet or dry deposition in the models can also partially explain the models' diversity. In addition, less constrained parameters like dissolution, long-range transport and Fe removal (which affect Fe solubility) can further increase the model diversity, and thus the STD, over these remote areas.

**5 Conclusions and future directions**

We here present the first model intercomparison study of the atmospheric Fe-cycle by assessing aerosol

simulations of total and labile Fe with four state-of-the-art global aerosol models (CAM4, IMPACT, GEOS-Chem and TM4-ECPL). The TFe emissions from dust sources in the models range from ~38 to ~134 Tg-Fe yr$^{-1}$, with a mean value of 71.5 ($\pm$43) Tg-Fe yr$^{-1}$. The models simulate the secondary formation of soluble Fe in the atmosphere, as a result of mineral Fe atmospheric processing by acids, organic ligands and photochemistry, but the absolute amount of the simulated LFe remains highly uncertain. The simulated LFe deposition fluxes from

mineral dust span from 0.3 to 1.0 Tg-Fe yr$^{-1}$, with a mean value of around 0.7 ($\pm$0.3) Tg-Fe yr$^{-1}$. All models capture the main features of the distribution of TFe and LFe: i.e., large deposition rates to the Sahara and Gobi Desert and regions downwind of strong dust sources. Models also show significant LFe deposition to the oceans downwind from the Middle East and the continents of South America, Africa and Australia in the Southern Hemisphere; the Middle East has large dust sources and while South America, Africa and Australia experience

strong biomass burning emissions.

The models are able to simulate the main features of TFe and LFe atmospheric concentrations and deposition fluxes. On average, the ensemble model computes roughly 50% higher lifetime with respect to the deposition flux for combustion aerosol (~14 days) than for dust aerosol, reflecting differences in size distribution and in the location of emitted aerosols. Ensemble model calculations present overestimate of the observed TFe surface mass

concentrations near the dust source regions and underestimate the Fe concentrations over the Southern Ocean compared with cruise measurements; similar to what it was pointed out in (Albani et al., 2014) and seen in the dust model intercomparison study of Huneeus et al. (2011). Note that the latter is important because of the key role of Fe in the biogeochemistry of these ocean waters. For the ensemble model mean, the MNB for the Northern Hemisphere (about 14) is larger than for the Southern Hemisphere (about 2.4). Note, however, that



evaluation of monthly mean model results by comparison with the shorter-term (e.g., daily) observations during different sampling periods introduces uncertainties.

The model intercomparison and model–observation comparison revealed the Fe size distribution and the relative contribution of the dust and combustion sources, as two critical issues for LFe simulations that now require

further study. The diversity of how the models represent Fe emissions as well as of deposition fluxes among the models can be thus large, especially over source regions. The model diversity over remote oceans reflects uncertainty in Fe content parameterizations of dust emissions (e.g., soil mineralogy and the initial Fe soluble content in primary sources) and combustion aerosols, and/or in the parameterizations of the size distribution of the transported aerosol Fe, and thus the representation of deposition fluxes - which overall control the

atmospheric lifetime of Fe. On the other hand, there are many other intrinsic reasons for this diversity especially for the LFe aerosol fraction, since it involves complex atmospheric chemical processes driven by atmospheric acidity. For example, detailed chemical mechanisms need to be invoked to simulate a multi-phase, multi-component solution system, since such a system may not be solved accurately using a thermodynamic equilibrium approach for the entire grid box due to sub-grid processes, e.g., when the dust plume is not well

mixed with surrounding pollutants. Consequently, a reasonable aerosol pH simulation further depends on the representation of soluble acidic and basic compounds, as well as the water content of hygroscopic particles.

In this respect, new field observations are needed to improve understanding of Fe solubilisation process, and how this process alters in the presence of anthropogenic pollution. Modelling studies together with their evaluation based on a greater number of atmospheric observations, especially over the remote ocean, are deemed necessary

in order to reduce the uncertainty associated to the model performance in simulating the atmospheric Fe deposition. For example, the participating models here predict that wet deposition processes are important for the LFe atmospheric cycle but, it is rather hard to test that well due to the lack of respective field data. Moreover, although the models do well in higher dust and pollution regions, from an oceanographer's perspective, the regions with the lowest Fe supply is of the greatest interest, because that creates the HNLC situation in the water

column. Model evaluation can be further more difficult, however, due to a lack of standardization in the protocols used to determine the soluble Fe fraction in marine aerosol samples (e.g., Baker and Croot, 2010). Protocols that involve different solutes, aerosol – solution contact times and filter pore sizes, among other differences, are in use by different investigators and these presumably introduce some, as yet unquantified, uncertainties into the available databases of aerosol soluble Fe concentrations and Fe solubility. Model developments related to

atmospheric Fe cycle must be performed in parallel with an extensive model evaluation in order to better



understand the underlying mechanisms and to provide, overall, realistic labile Fe deposition fluxes for the next-generation of ocean biogeochemistry modelling studies.



**Data availability:** Global 1.0x1.0 fields for the TFe and LFe deposition fluxes as well as the corresponding Fe solubility derived from the ensemble model calculations will be freely available after the final publication of this work (contact s.myriok@uu.nl).

**Author contribution:** This paper resulted from the deliberations of United Nations GESAMP Working Group
38, "The Atmospheric Input of Chemicals to the Ocean" (SM, AI, MK, AN, NM, AB, TJ, MS, YG, RS, MP, CG, RD), February 2017, University of East Anglia, Norwich, UK. SM manipulated all model fields, analyzed the data and prepared the respective figures and fields of this work. AI performed the model evaluation and the respective statistical analysis. JFK provided the analysis for dust correction factors for model data. SM, AI, MK, MCK, AN, NM, RAS, DSH, NM and MSJ, provided the model products, AB, TJ, MS, BS, YG, RS, CB, WL,
AB, TW, CG and YF contributed to the measured data. All authors contributed to the manuscript preparation.

**Acknowledgments:** SM acknowledges financial support for this research by the Marie-Curie H2020-MSCA-IF-2015 grant ODEON (ID 705652). Support for this research was provided to AI by JSPS KAKENHI Grant Number JP16K00530 and Integrated Research Program for Advancing Climate Models (MEXT). NM, RS and DH acknowledge support from DOE DE-SC0006791. We are grateful to T. Wagener for kindly providing field
data set. This work is prepared in the framework of the Joint Group of Experts on the Scientific Aspects of Marine Environmental Protection (GESAMP; http://www.gesamp.org/), Working Group 38, the Atmospheric Input of Chemicals to the Ocean. We thank the Global Atmosphere Watch and the World Weather Research Programme of the World Meteorological Organization, the International Maritime Organization, the U.S. National Science Foundation, the ICSU Scientific Committee on Oceanic Research (SCOR), and the University
of East Anglia for their financial support, and SOLAS for their sponsorship.

**Competing interests:** The authors declare that they have no conflict of interest.





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



**Tables**

**Table 1.** General description of the participating models used for the atmospheric Fe simulations. For multiple year simulations, the average was used.

| Model | Simulated year(s) | Horizontal resolution (lon x lat) | Vertical Resolution (sigma levels) | Meteorology | Reference |
|---|---|---|---|---|---|
| CAM4 | 2007-2011 | 1.9° x 2.5° | 56 | GEOS-5 | Scanza et al. (2018) |
| GEOS-Chem | 01/03/2009-28/02/2010 | 2.0° × 2.5° | 47 | GEOS-5 | Johnson and Meskhidze (2013) |
| IMPACT | 2014 | 2.0° × 2.5° | 59 | GEOS-FP | Ito et al. (2018) |
| TM4-ECPL | 2008 | 3.0° x 2.0° | 34 | ERA-Interim | Myriokefalitakis et al. (2015, 2016) |



**Table 2. Iron representation in the models; TFeD: total Fe in mineral dust; TFeC: total Fe in combustion aerosols Fe; LFeD: Labile Fe in mineral dust; LFeC: Labile Fe in combustion aerosols.**

| Model | Aerosol Fe Species | Aerosol Size Representation | Soil Mineralogy | Atmospheric Processing | Aqueous-phase median |
|---|---|---|---|---|---|
| CAM4 | TFeD, TFeC, LFeD, LFeC | 4 bins for each Fe-type with diameters: 0.1-1.0, 1.0-2.5, 2.5-5.0, and 5.0-10.0 µm | Claquin et al. (1999) | Proton- and ligand-dissolution of dust and combustion aerosols | Aerosol water, Cloud droplets |
| GEOS-Chem | TFeD, LFeD | 4 bins for TFeD with diameters: 0.2–2.0, 2.0–3.6, 3.6–6.0 and 6.0–12.0 µm 1 bulk for LFeD | Nickovic et al. (2012) | Proton- and ligand-dissolution of dust aerosols | Aerosol water, Cloud droplets |
| IMPACT | TFeD, TFeC, LFeD, LFeC | 4 bins for each Fe-type with diameters: <1.26, 1.26–2.5, 2.5–5, and 5–20 µm | Journet et al. (2014) | Proton-, ligand- and photoinduced dissolution of dust and combustion aerosols | Aerosol water |
| TM4-ECPL | TFeD, TFeC, LFeD, LFeC | 2 modes for each Fe-type, with mass median radii (lognormal standard deviation) of 0.34 µm (1.59) and 1.75 µm (2.00) for dust aerosols and 0.04 µm (1.8) and 0.5 µm (2.00) for combustion aerosols | Nickovic et al. (2012) | Proton- and ligand-dissolution of dust aerosols | Aerosol water, Cloud droplets |





**Table 3. Global mean bias correction factors (median, lower 95% confidence interval and upper 95% confidence interval) derived for each model and each aerosol size (bins or modes) based on state-of-the-art constraints on size-resolved dust loading (Kok et al., 2017), taken into account for the ensemble model calculations.**

| Model | AEROSOL SIZE | | | |
|---|---|---|---|---|
| | **Bin 1** | **Bin 2** | **Bin 3** | **Bin 4** |
| **CAM4** | 0.91, 0.55, 1.57 | 0.89, 0.57, 1.32 | 0.98, 0.62, 1.46 | 1.60, 0.97, 2.44 |
| **GEOS-Chem** | 1.37, 0.86, 2.09 | 0.99, 0.63, 1.44 | 0.91, 0.56, 1.38 | 0.24, 0.76, 1.94 |
| **IMPACT** | 0.89, 0.54, 1.38 | 0.71, 0.45, 1.01 | 0.91, 0.564, 1.30 | 1.01, 0.62, 1.49 |
| **TM4-ECPL***| **Accumulation Mode** | | **Coarse Mode** | |
| | 0.37, 0.22, 0.57 | | 0.73, 0.43, 1.05 | |

*The median correction factors for the corresponding bins 1-4 are 0.134, 0.692, 1.257 and 1.81, respectively (see text).

**Table 4. Annual budgets (Tg-Fe yr⁻¹) of total (TFe) and labile (LFe) Fe for emissions (EMI), dry deposition (DRY), wet deposition (WET) and sources (SRC; i.e., sum of emissions and atmospheric processing) for dust (TFeD, LFeD) and combustion (TFeC, LFeC) Fe-containing aerosols as calculated by the models, as well as the models' mean (± standard deviation; STD).**

| Model | TFeD | | | TFeC | | | LFeD | | | LFeC | | |
|---|---|---|---|---|---|---|---|---|---|---|---|---|
| | **EMI** | **DRY** | **WET** | **EMI** | **DRY** | **WET** | **SRC** | **DRY** | **WET** | **SRC** | **DRY** | **WET** |
| **CAM4** | 57.1 | 33.4 | 23.7 | 1.9 | 0.8 | 1.1 | 1.0 | 0.2 | 0.8 | 0.1 | 0.01 | 0.1 |
| **GEOS-Chem** | 56.5 | 40.2 | 16.3 | - | - | - | 0.8 | 0.6 | 0.2 | -- | - | - |
| **IMPACT** | 134.1 | 67.9 | 66.2 | 2.7 | 1.0 | 1.8 | 0.7 | 0.3 | 0.4 | 0.1 | 0.01 | 0.1 |
| **TM4-ECPL** | 38.1 | 30.34 | 7.76 | 1.8 | 1.4 | 0.4 | 0.3 | 0.1 | 0.2 | 0.2 | 0.1 | 0.1 |
| **MEAN (± STD)** | 71.5 (± 42.69) | 43.0 (± 17.13) | 28.5 (± 26.97) | 2.1 (± 0.51) | 1.1 (± 0.35) | 1.1 (± 0.71) | 0.7 (± 0.28) | 0.3 (±0.20) | 0.4 (±0.28) | 0.1 (± 0.05) | 0.1 (± 0.06) | 0.1 (± 0.01) |



**Table 5. Annual deposition fluxes (Tg-Fe yr$^{-1}$) to different ocean basins of total (TFe) and labile (LFe) Fe, as calculated by the contributing models and the derived ENSEMBLE model.**

| Ocean Basin | CAM4 | | GEOS-Chem[*] | | IMPACT | | TM4-ECPL | | ENSEMBLE | |
|---|---|---|---|---|---|---|---|---|---|---|
| | TFe | LFe | TFe | LFe | TFe | LFe | TFe | LFe | TFe | LFe |
| North Atlantic | 4.914 | 0.202 | 6.407 | 0.109 | 7.541 | 0.065 | 3.903 | 0.057 | 5.07 | 0.096 |
| South Atlantic | 0.681 | 0.043 | 2.067 | 0.017 | 4.414 | 0.014 | 0.455 | 0.012 | 1.677 | 0.02 |
| North Pacific | 1.084 | 0.057 | 1.653 | 0.041 | 0.972 | 0.021 | 3.075 | 0.054 | 1.496 | 0.038 |
| South Pacific | 0.111 | 0.007 | 1.114 | 0.01 | 2.337 | 0.014 | 0.49 | 0.012 | 0.867 | 0.009 |
| Indian | 3.658 | 0.076 | 3.818 | 0.049 | 6.911 | 0.09 | 1.33 | 0.027 | 3.644 | 0.055 |
| Mediterranean | 1.596 | 0.019 | 2.664 | 0.043 | 4.005 | 0.021 | 0.234 | 0.004 | 1.95 | 0.018 |
| Baltic Sea | 0.011 | 0.001 | 0.011 | <0.001 | 0.017 | <0.001 | 0.014 | <0.001 | 0.011 | <0.001 |
| Black & Caspian Sea | 0.342 | 0.006 | 0.398 | 0.005 | 0.966 | 0.009 | 0.15 | 0.001 | 0.439 | 0.005 |
| Hudson Bay | 0.003 | <0.001 | 0.006 | <0.001 | 0.003 | <0.001 | 0.018 | <0.001 | 0.007 | <0.001 |
| Arctic (>66N) | 0.127 | 0.008 | 0.142 | 0.001 | 0.063 | 0.001 | 0.411 | 0.004 | 0.162 | 0.003 |
| Southern (>60S) | 0.010 | 0.001 | 0.030 | <0.001 | 0.058 | <0.001 | 0.051 | <0.001 | 0.031 | <0.001 |
| Global Ocean | 12.538 | 0.419 | 18.31 | 0.275 | 27.287 | 0.235 | 10.131 | 0.173 | 15.354 | 0.246 |

10  [*]In GEOS-Chem, only the Fe from mineral dust is considered





**Figures**

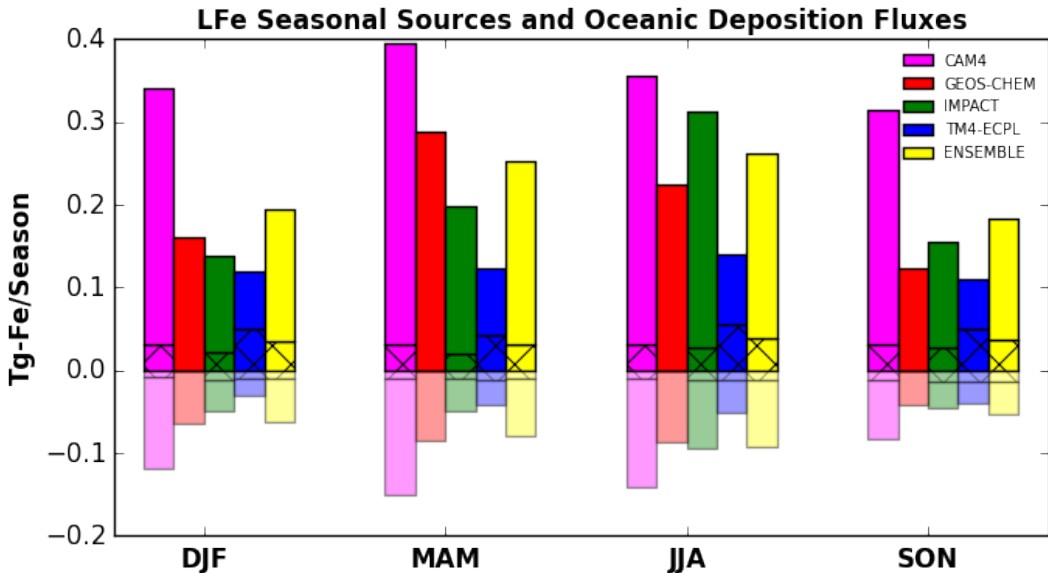

**Figure 1: Seasonal LFe sources (positive bars) and oceanic deposition fluxes (negative bars/pale colors) in Tg-Fe Season[-1] for December, January and February (DJF); March, April and May (MAM); June, July and August (JJA) and September, October and November (SON), as calculated by each model (CAM4: magenta; GEOS-Chem: red; IMPACT: green and TM4-ECPL: blue), as well as, the ENSEMBLE model (yellow). The hatched areas correspond to the combustion aerosols.**





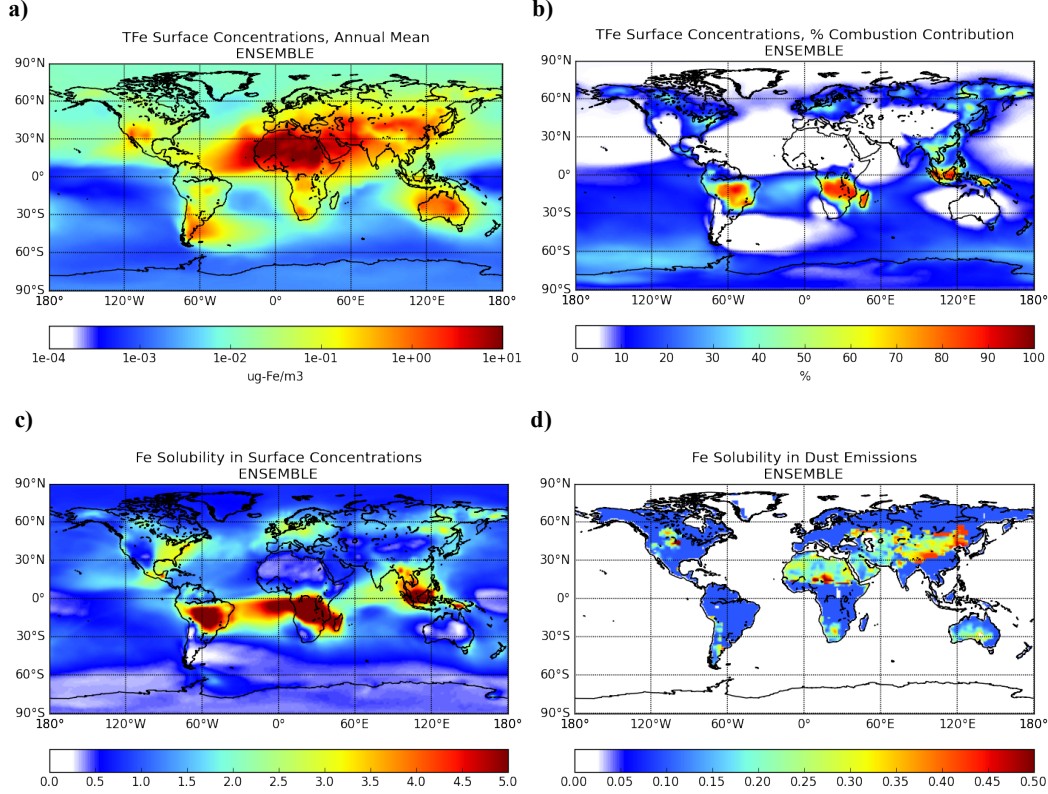

**Figure 2: ENSEMBLE model results for annual mean (a) surface TFe concentration (μg m⁻³), (b) the percentage contribution (%) of Fe-containing dust aerosols, (c) the Fe solubility (%) in surface TFe concentration and (d) the initial solubility (%) in Fe-containing dust emissions.**



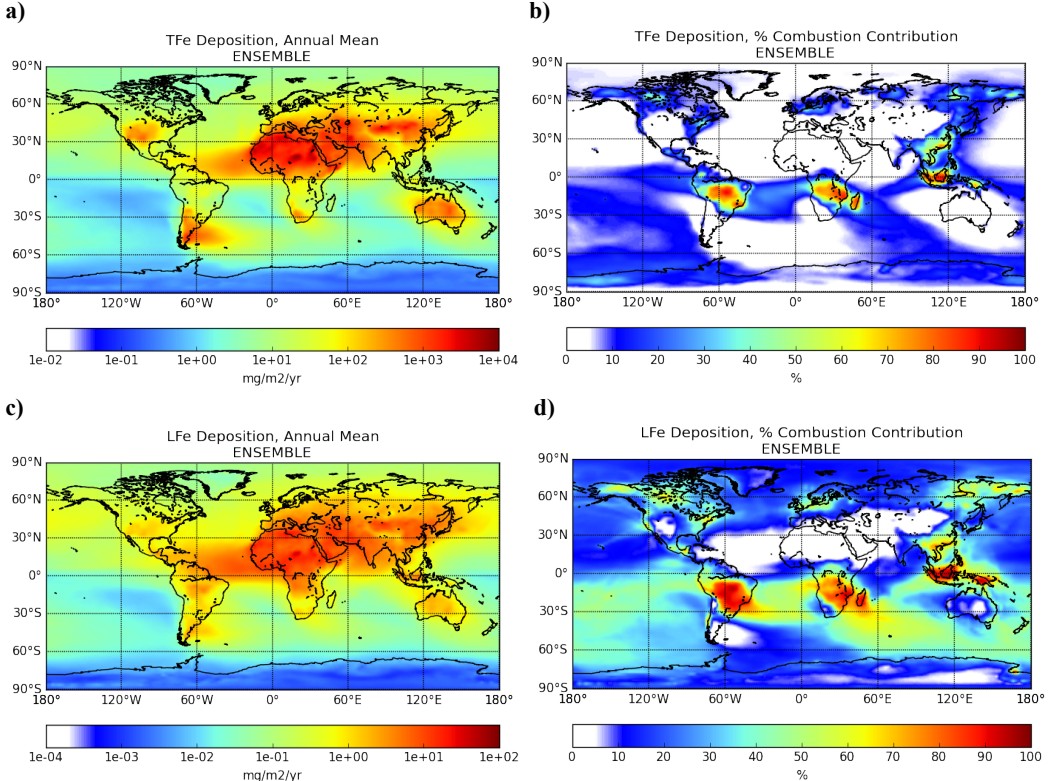

**Figure 3: ENSEMBLE model results for annual deposition fluxes (mg-Fe m$^{-2}$ yr$^{-1}$) for (a) TFe and for (c) LFe and their respective percentage contribution (%) of combustion aerosols (b, d).**





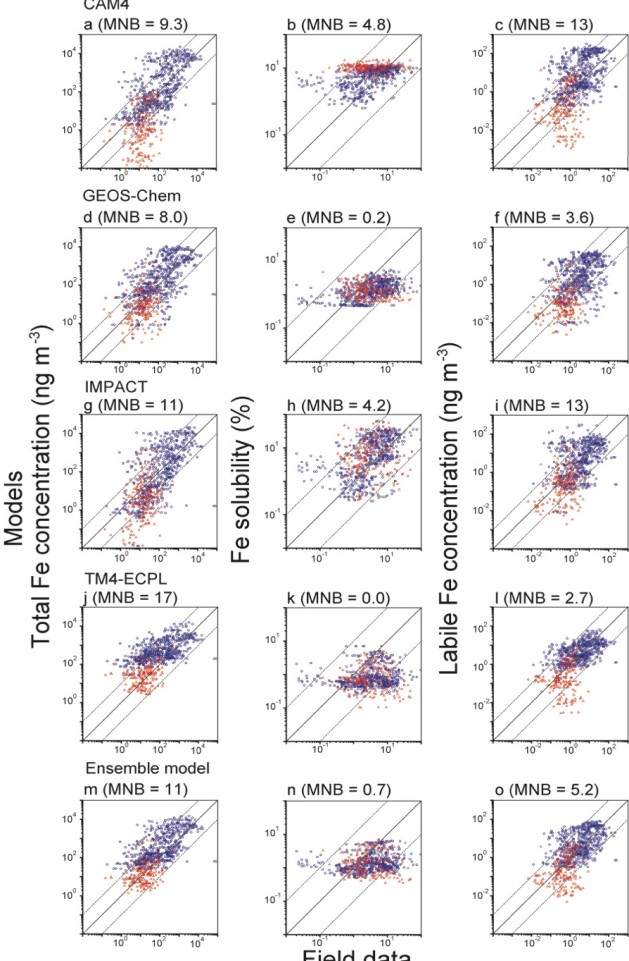

**Figure 4: Comparison of simulated and observed TFe loading (ng m$^{-3}$), Fe solubility (%), and LFe loading (ng m$^{-3}$) in the Northern (blue circles) and Southern (red squares) Hemisphere. a, b, and c CAM4; d, e, and f GEOS-Chem; g, h, and i IMPACT; j, k, and l TM4-ECPL; m, n, and o for the ENSEMBLE model. The mean normalized biases (MNB) between the models and observations are presented in parentheses. The solid line represents a 1:1 correspondence and the dashed lines show the 10:1 and 1:10 relationships, respectively. The bias correction in the mineral dust size distribution is applied for the comparison with field data (Kok et al., 2017).**





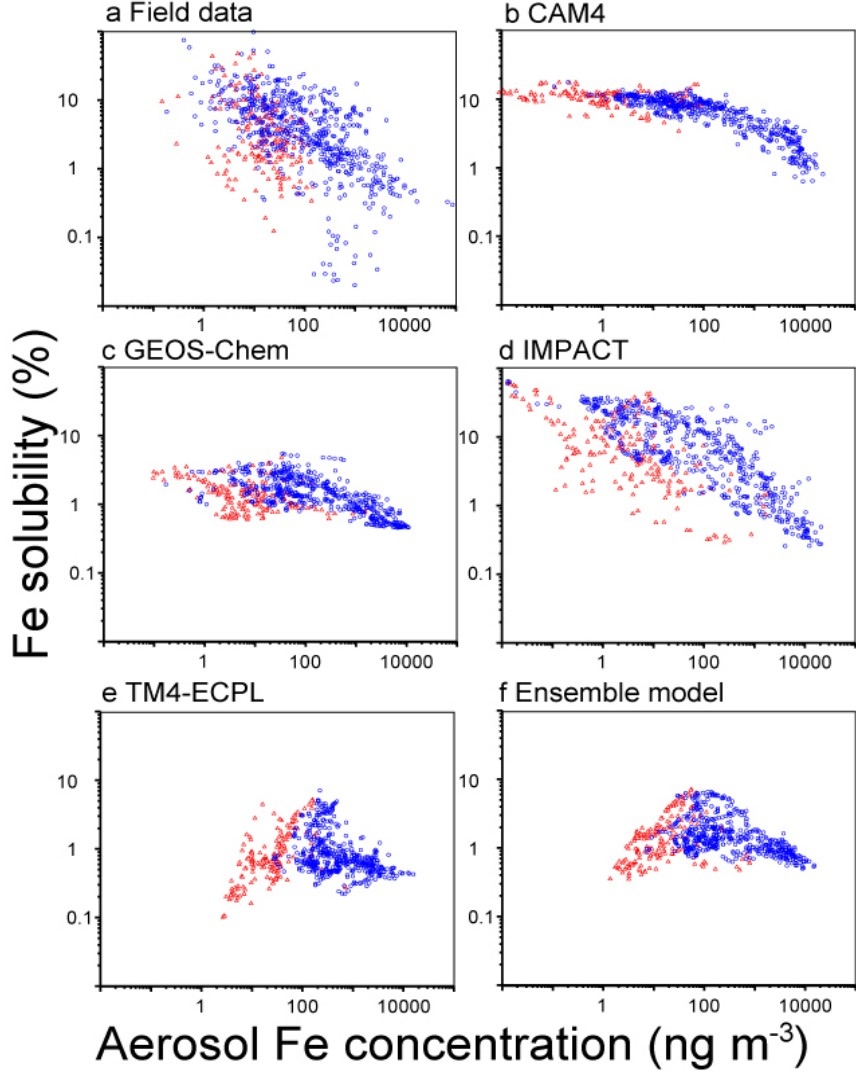

**Figure 5: Fe solubility versus atmospheric loading of aerosol Fe (ng m⁻³) in the Northern (blue circles) and Southern (red squares) Hemisphere for (a) the measurements, (b) CAM4, (c) GEOS-Chem (d) IMPACT, (e) TM4-ECPL and (f) the ENSEMBLE model. The bias correction in the mineral dust size distribution is applied for the comparison with field data (Kok et al., 2017).**





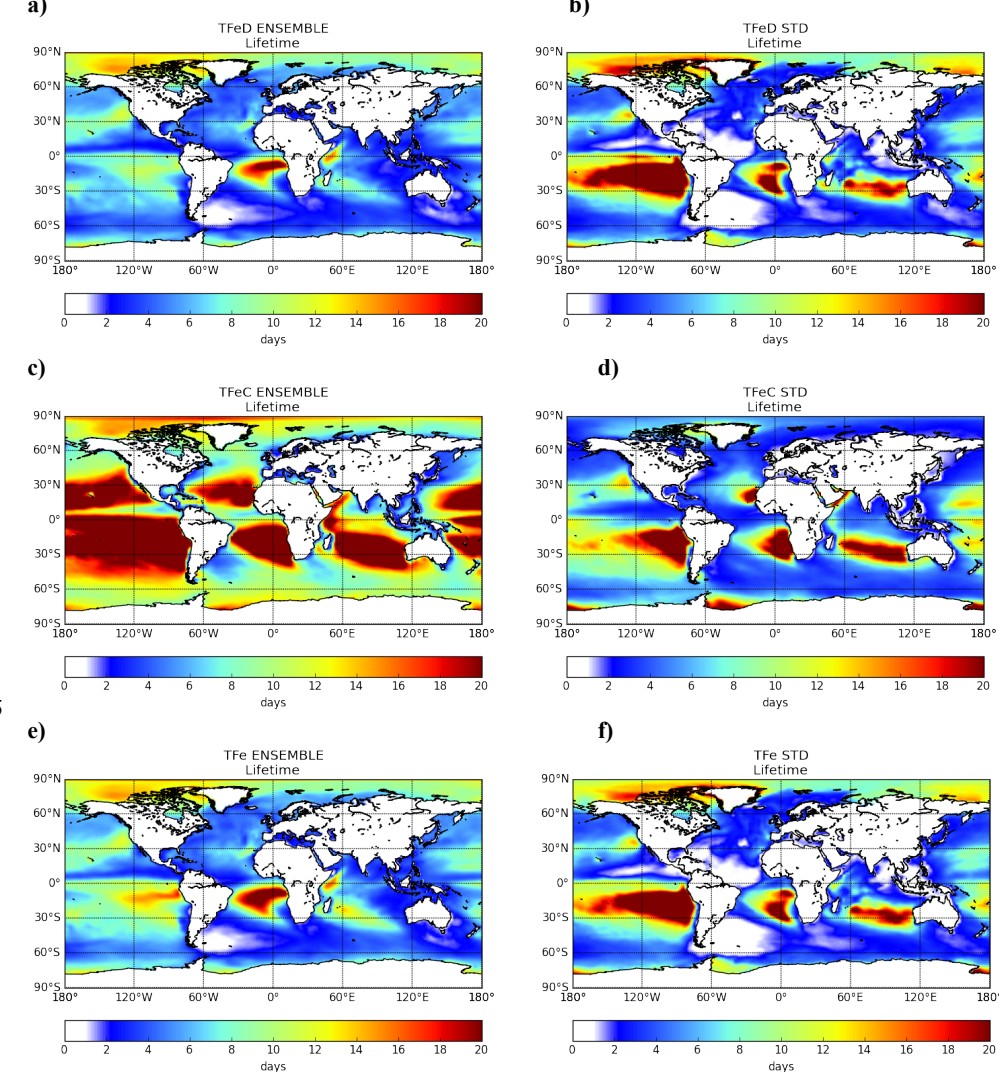

**Figure 6: ENSEMBLE model results for TFe lifetime (days) over the ocean originated from a) mineral dust sources c) combustion source and e) total (mineral dust + combustion) and the respective standard deviation (b,d,f). Lifetimes are burdens divided by total sinks.**