# Peer review of "The GESAMP atmospheric iron deposition model intercomparison study"

_Biogeosciences, 2018_

## Referee Comment (RC1) · Anonymous Referee #1 · 31 Jul 2018

Biogeosciences Title: The GESAMP atmospheric iron deposition model intercomparison study Author(s): Stelios Myriokefalitakis et al. MS No.: bg-2018-285 MS Type: Reviews and syntheses

This is clearly interesting and worthwhile work, given the importance of iron in ocean nutrition, but I think there is an immediate question about why it is needed. The authors state that this kind of modelling is the only way to estimate iron deposition (page 4 lines 11ff). But is it really impossible to utilise the large number of observations of Fe concentrations (listed in the SI) to estimate fluxes? I would like to see a better justification for this claim.

[Figure]

The third aim of the work (page 7, top) seems circular – why would future modelling studies find the fluxes calculated in this modelling study useful, other than as comparative measures? I also miss an indication that the work described here is potentially useful in permitting prediction of changes in Fe deposition rates, for example due to anthropogenic activities.

Page 21, line 10 states "The TFe loading, Fe solubility, and LFe loading from the models are compared with the measurements and presented in Fig. 4." This confuses me, since I would use the term loading to mean a flux over time (mass per unit area). The axis labels refer to concentrations, with units of mass per volume, but the text and the Figure caption use loading. Please clarify.

And also with reference to Figure 4, if I understand correctly (and if I don't then please clarify the text) the MNB values indicate the overall bias of the predictions compared to the data, which would mean that the ensemble model overestimates LFe concentrations by a factor of five. Does it then follow that loadings to the ocean are overestimated by this factor? If so, then the proposed further work doesn't seem to address the issue – Section 5 reads more like a series of minor tweaks than addressing a major quantification problem.

Evaluating the importance of atmospherically deposited Fe depends greatly upon assessing the fate of the metal in ocean water. According to the authors "Upon deposition to the surface ocean, this fraction of Fe from the atmosphere can either enter the dissolved Fe pool, or precipitate-out as large oxy-hydroxide particles (Meskhidze et al., 2017)". I am surprised that the cited study, which worked with high Fe concentrations and did not explore the influence of light on iron chemistry, is considered to represent the state of knowledge in this area. I am also surprised that neither this reference nor the paper under review cites the book by Turner and Hudson "The Biogeochemistry of Iron in Seawater" (Wiley 2001).

Is there any prospect of using the Fe loadings reported here to simulate Fe concentra-

tions in the ocean? I realise that this may be outside the scope of the present paper, but some indication of possibilities would be welcome.

It is not clear to me whether FeD deposited to the ocean is considered "inert" or whether it can yield significant dissolved Fe. Maybe this could be explained. If it is not considered to be a source, then it is not so important to get the global fluxes correct, and the focus should be on the LFe.

As I understand it, a similar loading (to LFe) of dissolved Fe to the oceans comes from rivers. Could the authors briefly explain why this is not considered as important as the atmospherically-deposited form?

Section 2.1.2 introduces the presence of oxalate in aerosols, without explanation of its sources and why other carboxylic acids are not considered. I am not at all expert in this area, it appears as though oxalate is assumed or known to be dominant – if so then its strong solubilising properties are clearly important. I would appreciate some references to justify the assumption that oxalate is truly dominant in governing aerosol Fe solubility.

The right hand maps in Figure S4 are not informative. Is it possible – or do the authors consider it worthwhile? – to show primary sources of LFe?

TECHNICAL CORRECTIONS (just the ones I noted, this is not comprehensive!) Page 5 line 15 This sentence needs improvement. Table S3. What does "NaN" mean? (not analysed I guess, but please say). Figure S7 should have "continuous" not "continues" Page 21 line 19. I assume that SH = southern hemisphere? Is this so very well known? Page 21 line 22. Should it read 0.50-0.56?

---

## Referee Comment (RC2) · Anonymous Referee #2 · 21 Aug 2018

Review Myriokefalitakis et al

The authors present here an intercomparison of four state-of-the-art global models of atmospheric Fe concentrations and deposition. Atmospheric Fe from both mineral dust and combustion processes is considered and the intercomparison deals with total Fe and labile Fe. In addition to the model intercomparison, the authors also compare model results with observations. This is a very nice and interesting piece of work. The manuscript is well written and it deserves publication in Biogeosciences. My main concern is about the interannual variability of the models. As stated in Table 1, the simulated years are different for each model but the interannual variability of each model

is not presented nor discussed. Moreover, no requests for meteorological conditions or emission inventories have been set to the model simulations and the sensitivity of each model to these parameters are also not discussed.

Few additional minor comments are listed below.

P2, line 1: please add min and max for TFe and LFe deposition fluxes.

P3, lines 18-22: the fraction of Fe that is bioavailable is still not well known and also depends on phytoplankton species, so I suggest that the authors do not write that labile Fe is a good approximation for bioavailable Fe.

P4, line 8: "can be also be"

P6, lines 2-5: for the role of oxalate on Fe solubility, Paris et al. (2011) could be cited as well (https://doi.org/10.1016/j.atmosenv.2011.08.068).

P7, line 29: please change "in (Albani et al., 2014)" by "in Albani et al. (2014)"

P18, lines 28-30: "LFe sources are mainly driven by mineral dust aerosols, although a significant fraction (6 to 62%) is due to LFe combustion aerosols, especially over the high-latitudes of the Northern Hemisphere (Ito et al. 2018; companion manuscript to be submitted)." I would rather put this sentence in the previous section.

P19, lines 1-5: the authors compare the seasonal variability of LFe, but I would have liked to see the error bars on Fig. 1, as well as more information on the statistical test (which one was used, P value, n,...). Moreover, the authors state that "in most of the cases IMPACT and GEOS-Chem present similar seasonal variation." However, IMPACT is higher in JJA, while GEOS is higher in MAM.

P19, lines 9-11: the authors state that "in the other seasons the 30N maximum is not clearly present", but in JJA, a clear maximum for IMPACT is seen at 30°N.

P21, line 12: the authors should explain how they calculate the mean normalized bias. Why would a value of 2.4 mean that the concentrations are underestimated? This is

not clear to me.

P22, line 3 and Fig. 5: the authors discuss the relationship between Fe solubility and aerosol Fe concentrations but these 2 variables are not independent as the latter one is used to calculate the former one. How do the authors deal with that?

P23, line 12-: How is the lifetime (turnover time) calculated? Is it calculated by dividing the concentration by the deposition flux, both estimated by the model? This could be added in the text.

P24, lines 26-27: please change "similar to what it was pointed out in (Albani et al., 2014) and seen in the dust model intercomparison study of Huneeus et al. (2011)." By "similar to what was pointed out in Albani et al. (2014) and seen in the dust model intercomparison study of Huneeus et al. (2011)".
* * *

---

## Author Comment (AC1) · 10 Sep 2018

We thank the Referee#1 for the careful reading of the paper. Please find below the point-by-point answers to the referee's general and specific comments and technical corrections.

1. General comments:

Q: The authors state that this kind of modelling is the only way to estimate iron deposition (page 4 lines 11ff). But is it really impossible to utilise the large number of observations of Fe concentrations (listed in the SI) to estimate fluxes? I would like to

see a better justification for this claim.

A: With our statement we do not want to devaluate the importance and the need of observations. We state that models are an excellent way to study atmospheric Fe supply to the oceans and to assess its impacts on a global scale, partly because of episodic nature of atmospheric deposition. Models enable the integration of knowledge and of discontinuous geographically and temporally observations and in synergy with observations are the appropriate tools to study the global spatial and temporal patterns of species such as Fe. We rephrased as: "The use of global biogeochemical numerical models and surface observations is an excellent way to better understand past, present and future atmospheric supply to the oceans, as well as to quantify the resultant effect on the ocean biological productivity and the carbon uptake."

Q: The third aim of the work (page 7, top) seems circular – why would future modelling studies find the fluxes calculated in this modelling study useful, other than as comparative measures? I also miss an indication that the work described here is potentially useful in permitting prediction of changes in Fe deposition rates, for example due to anthropogenic activities.

A: The 3rd aim refers to the utility of the calculated ensemble Fe deposition as an input for the next-generation of ocean biogeochemistry modelling studies. Our study aims to provide to the scientific community with ensemble TFe and LFe deposition fluxes, as a result of state-of-the-art atmospheric models and satellite retrievals. Currently, most ocean biogeochemistry models use global dust deposition fields to derive the atmospheric Fe input (e.g., Aumont et al., 2015), usually by assuming a constant fraction by mass on dust. Furthermore, to take into account the labile fraction in Fe deposition fluxes, either a constant value is applied or Fe solubility maps from other atmospheric models are used (e.g., Mahowald et al., 2005). Such approaches mean that the ocean data will contain significant assumptions, which the current science no longer supports; e.g., the combustion Fe-containing aerosol and the heterogeneity of soil minerology are neglected as well as the explicitly calculation of the Fe solubilization processes (resulting in lower Fe solubility near the dust source regions and higher Fe solubility over the remote ocean than earlier estimates, bringing closer model results to observations). Overall, models need to first be evaluated against observations though to have some level of confidence in their ability and then to use them in order to access for example the anthropogenic effects. We can further use the models for the predicting of changes in Fe deposition rates, especially for the past and the future.

2. Specific comments:

Q: Page 21, line 10 states "The TFe loading, Fe solubility, and LFe loading from the models are compared with the measurements and presented in Fig. 4." This confuses me, since I would use the term loading to mean a flux over time (mass per unit area). The axis labels refer to concentrations, with units of mass per volume, but the text and the Figure caption use loading. Please clarify.

A: We changed the term "loading" to "concentrations".

Q: With reference to Figure 4, if I understand correctly (and if I don't then please clarify the text) the MNB values indicate the overall bias of the predictions compared to the data, which would mean that the ensemble model overestimates LFe concentrations by a factor of five. Does it then follow that loadings to the ocean are overestimated by this factor? If so, then the proposed further work doesn't seem to address the issue – Section 5 reads more like a series of minor tweaks than addressing a major quantification problem.

A: As for the MNB values, overestimates are weighted more than equivalent underestimates. As was noted in p.24, l.24, a similar overestimate in the measured monthly averaged dust concentration from a the short-term cruise measurements was seen in the dust model intercomparison study of Huneeus et al. (2011). The bias may be due to the short duration of the sampling frequencies. We added further work to address one of the major quantification problems in p.25, l.2, "due to to short of the sampling frequencies. A comparison of long-term measurements with a multi-year modelling

will allow assessment of the model performance to capture labile Fe concentrations under specific events" after "Note, however, that evaluation of monthly mean model results by comparison with the shorter-term (e.g., daily) observations during different sampling periods introduces uncertainties". We also added the description of MNB in the text as following: "We use the monthly mean of model output to compare with the measurements. The normalized bias (NB) at a given grid box is calculated as follows: NBâŰă(@i)=(CâŰă(@model,i)-CâŰă(@obs,i))/(CâŰă(@obs,i)) (2) where, Cmodel,i is the modelled aerosol concentration in grid box i , and Cobs,i is the measured aerosol concentration in the same grid box. When discussing the multi-model results we use the mean of all models, while we also analyze the mean normalized bias (MNB) of the models against measurements (a perfect comparison would have a MNB of 0 and cor-relation, R, of 1). A model's MNB is derived as the arithmetic mean of all NBi values, thus overestimates are weighted more than equivalent underestimates."

Q: Evaluating the importance of atmospherically deposited Fe depends greatly upon assessing the fate of the metal in ocean water. According to the authors "Upon depo-sition to the surface ocean, this fraction of Fe from the atmosphere can either enter the dissolved Fe pool, or precipitate-out as large oxy-hydroxide particles (Meskhidze et al., 2017)". I am surprised that the cited study, which worked with high Fe concentrations and did not explore the influence of light on iron chemistry, is considered to represent the state of knowledge in this area. I am also surprised that neither this reference nor the paper under review cites the book by Turner and Hudson "The Biogeochemistry of Iron in Seawater" (Wiley 2001).

A: The study by Meskhidze et al. (2017) was designed to represent the processes af-fecting the soluble Fe deposited to the open oceans through atmospheric pathways on a time scale of sec to minutes. The concentrations were selected to be representative of wet removal (i.e., rainout and washout) which is the dominant removal mechanism over the remote oceans. The reference is used here, because, as far as we know, this is the first study that explored the role of atmospheric organic ligands on Fe solubility

after deposition to the surface ocean. However, we agree with the reviewer tha additional references, particularly on the effects of oceanic ligands and photochemistry need to be provided. The revised manuscript now reads: "Upon deposition to the surface ocean, this fraction of Fe from the atmosphere can either enter the dissolved Fe pool or precipitate-out as large oxy-hydroxide particles (de Baar and de Jong, 2001; Boyd and Ellwood, 2010; Meskhidze et al., 2017; Turner and Hunter, 2001)."

Q: Is there any prospect of using the Fe loadings reported here to simulate Fe concentrations in the ocean? I realize that this may be outside the scope of the present paper, but some indication of possibilities would be welcome.

A: We included the following part in the conclusions: Although the calculation of the oceanic Fe concentrations is outside the scope of this paper, we do hope that the deposition fields provided by this work will be used for this purpose, since they will trigger such investigation to occur.

Q: It is not clear to me whether FeD deposited to the ocean is considered "inert" or whether it can yield significant dissolved. Maybe this could be explained. If it is not considered to be a source, then it is not so important to get the global fluxes correct, and the focus should be on the LFe.

A: Total Fe deposited in the ocean is important for the assessment of the fate of Fe in the ocean. Total Fe is needed for the comparison of particulate Fe with the measurements in the ocean biogeochemistry models (e.g., Ye and Völker, 2017). Additionally, less labile Fe in total Fe may be potentially utilized by marine organisms. Note that ocean biogeochemistry models (e.g., Aumont et al., 2015) take into account both the total and the soluble deposited Fe for chemistry calculations, assuming some fractions of less labile Fe in total Fe are dissolved in the ocean. For example, Aumont et al. (2015) considers that the particulate Fe from dust experiences dissolution in the water column, with the dissolution rate computed assuming that during sinking of mineral particles particulate Fe dissolves by about 0.01% per day (Bonnet, 2004). Therefore,

both the total and the labile Fe deposition fluxes are needed. However, the dissolution of Fe from FeD is species depended and affected by spatiotemporal variations in the ocean. The following explanation has been added in the text in the introduction: "Both the TFe and LFe atmospheric deposition can be used in ocean biogeochemical modelling. For example, total Fe is needed for comparisons of particulate Fe with the measurements in the ocean biogeochemistry models (e.g., Ye and Völker, 2017), while LFe can be assumed as readily available to the marine ecosystem. Note that the less labile fraction of Fe in TFe can be slowly dissolved from particulate Fe in the ocean during sinking of mineral particles (e.g., roughly 0.01% per day; Bonnet, 2004), with the dissolution of Fe, however being species depended and affected by spatiotemporal variations in the ocean"

Q: As I understand it, a similar loading (to LFe) of dissolved Fe to the oceans comes from rivers. Could the authors briefly explain why this is not considered as important as the atmospherically-deposited form?

A: According to the recent study of Tagliabue et al. (2016), riverine inputs are considering 1-2 order of magnitudes smaller than the atmospheric dust deposition to the global ocean. We incluied the following part in the manuscript (introduction), to further refer to the other known sources of Fe in the global ocean: "However, significant Fe inputs from continental margins and hydrothermal vents are also supplied to the global ocean, regulating the ocean biogeochemical cycles. Moreover, riverine Fe inputs are currently estimated 1-2 orders of magnitudes smaller that the atmospheric pathway (e.g., Tagliabue et al., 2016), affecting mainly coastal regions, while icebergs and glaciers could also be important to the polar oceans (Raiswell et al., 2016)."

Q: Section 2.1.2 introduces the presence of oxalate in aerosols, without explanation of its sources and why other carboxylic acids are not considered. I am not at all expert in this area, it appears as though oxalate is assumed or known to be dominant – if so then its strong solubilising properties are clearly important. I would appreciate some references to justify the assumption that oxalate is truly dominant in governing aerosol

Fe solubility.

A: Indeed, numerous organic compounds, such as acetate, formate, oxalate, malonate, succinate, glutarate, glycolate, lactate, tartrate and humic like substances (HULIS) can be found in atmospheric waters. However, oxalate, malonate, tartrate and humic acid have been observed to enhance Fe solubility (e.g., Paris et al., 2010, 2011). For all these organic ligands, positive dependences of iron solubility to organic concentrations were observed and revealed that the extent of organic complexation on iron solubility decreased in the following order: oxalate > malonate = tartrate > humic acid (Paris et al., 2011). Therefore, this study confirmed that among the known atmospheric organic binding ligands of Fe, oxalate is the most effective ligand in promoting dust iron solubility under atmospheric conditions. Furthermore, observations in the atmosphere, point to oxalate as the most abundant organic ligand (e.g., Kawamura and Ikushima, 1993; Kawamura and Sakaguchi, 1999). Oxalate originates from multiphase chemistry of organics, but has also weak anthropogenic primary sources (see Myriokefalitakis et al., 2011 and Lin et al., 2014 for a comprehensive global modelling study of atmospheric oxalate). Therefore, atmospheric models use oxalate to study the effect of organic ligands on Fe dissolution. However, the lack of experimental data for Fe-containing minerals mixed with a variety of organic ligands in solution is an important source of uncertainty. For clarity, in Sect. 2.1.2 we added after the first sentence the following explanatory text: "Oxalate is, however, used in models as a proxy of all organic ligands for ligand-promoted dissolution since 1) it is the most abundant in the atmosphere (e.g., Kawamura and Ikushima, 1993; Kawamura and Sakaguchi, 1999) originating mainly from secondary sources and only a weak contribution from combustion primary sources (e.g., Myriokefalitakis et al., 2011) and 2) it is the most effective ligand in promoting iron solubilisation (e.g., Paris et al., 2011). We note, however, that more work is required to elucidate the role of other ligands that may promote Fe dissolution in future studies.".

Q: The right-hand maps in Figure S4 are not informative. Is it possible – or do the

authors consider it worthwhile? – to show primary sources of LFe?

A: As we state in the text, not all the models simulate the LFe primary and secondary sources in the same manner of dust and combustion aerosols. For a fairer comparison in Fig. S4, we show the primary (i.e., emissions) and secondary (i.e., atmospheric processing) sources together. As we state in the manuscript "the models use significantly different assumptions to describe the total LFe source to the atmosphere and therefore primary (emissions) and secondary (atmospheric processing) sources cannot be accurately separated" for all models. A detailed description of models' parameterizations as well as the differences among them with regard to the LFe sources, are also presented in Sect. 2.1.

3. Technical Corrections:

Q: Page 5 line 15 This sentence needs improvement.

A: We rephrased the text between lines 13-19 as follows: "During atmospheric transport coating of Fe-containing dust particles by acidic compounds (e.g., sulfates and nitrates) increase the Fe solubility. When this process is taken into account in model simulations (e.g., Meskhidze et al., 2005) it aids in explaining the observations. Indeed, measurements of the fresh dust particles present low («1%) initial solubilities (Chuang et al., 2005; Fung et al., 2000; Hand et al., 2004; Sedwick et al., 2007), while high aerosol solubilities are commonly observed at lower dust concentrations far from sources (Baker and Jickells, 2006; Sholkovitz et al., 2012; Oakes et al., 2012). Atmospheric processing of dust (Kumar et al., 2010; Meskhidze et al., 2003; Srinivas et al., 2014) is considered as the best candidate to explain these observations."

Q: Table S3. What does "NaN" mean? (not analysed I guess, but please say).

A: NaN is replaced with "-", which means that data are not available. We have also modified and explain it now in the Table S3 caption.

Q: Figure S7 should have "continuous" not "continues".

A: Done

Q: Page 21 line 19. I assume that SH = southern hemisphere? Is this so very well known?

A: We replaced "SH" with "the Southern Hemisphere".

Q: Page 21 line 22. Should it read 0.50-0.56?

A: The value is correct. The differences in Fe solubility trend between CAM4 and TM4-ECPL can be partly seen from Fig. 5.

4. References

Aumont, O., Ethé, C., Tagliabue, A., Bopp, L. and Gehlen, M.: PISCES-v2: an ocean biogeochemical model for carbon and ecosystem studies, Geosci. Model Dev., 8(8), 2465–2513, doi:10.5194/gmd-8-2465-2015, 2015. de Baar, H. J. and de Jong, J. T.: 'Distributions, sources and sinks of iron in seawater, '., 2001. Baker, A. R. and Jick-ells, T. D.: Mineral particle size as a control on aerosol iron solubility, Geophys. Res. Lett., 33(17), L17608, doi:10.1029/2006GL026557, 2006. Baker, A. R., French, M. and Linge, K. L.: Trends in aerosol nutrient solubility along a west–east transect of the Sa-haran dust plume, Geophys. Res. Lett., 33(7), L07805, doi:10.1029/2005GL024764, 2006. Bonnet, S.: Dissolution of atmospheric iron in seawater, Geophys. Res. Lett., 31(3), L03303, doi:10.1029/2003GL018423, 2004. Boyd, P. W. and Ellwood, M. J.: The biogeochemical cycle of iron in the ocean, Nat. Geosci., 3(10), 675–682, doi:10.1038/ngeo964, 2010. Chuang, P. Y., Duvall, R. M., Shafer, M. M. and Schauer, J. J.: The origin of water soluble particulate iron in the Asian atmospheric outflow, Geophys. Res. Lett., 32(7), L07813, doi:10.1029/2004GL021946, 2005. Fung, I. Y., Meyn, S. K., Tegen, I., Doney, S. C., John, J. G. and Bishop, J. K. B.: Iron sup-ply and demand in the upper ocean, Global Biogeochem. Cycles, 14(1), 281–295, doi:10.1029/1999GB900059, 2000. Hand, J. L., Mahowald, N. M., Chen, Y., Siefert, R. L., Luo, C., Subramaniam, A. and Fung, I.: Estimates of atmospheric-processed

[revised manuscript text omitted]

――――――――――――――――――

---

## Author Comment (AC2) · 10 Sep 2018

We thank the Referee#2 for the careful reading of the manuscript. Please find below the point-by-point answers to the referee's general, specific and technical comments.

1. General comments:

Q: My main concern is about the interannual variability of the models. As stated in Table 1, the simulated years are different for each model but the interannual variability of each model is not presented nor discussed.

A: This work uses single year model results only. Therefore, no year-to-year variability

is possible to be presented in our post-processed analysis for the present atmosphere. However, our analysis shows the variability derived from the structural differences between models.

Q: Moreover, no requests for meteorological conditions or emission inventories have been set to the model simulations and the sensitivity of each model to these parameters are also not discussed.

A: The aim of this work is to describe the current state of Fe global atmospheric deposition modeling and provide a multi-model ensemble of Fe atmospheric deposition fluxes of the current estimates, characterized with regard to observations (see introduction). It aims also to understand the origin of the respective model differences over the oceanic regions among the participating models and not to conclude which model is the best. As discussed in the model description (section 2), the participating models use different parameterizations to simulate the Fe-cycle in the atmosphere. Although using the same meteorology and the same emission inventories in the participating models would have been a very interesting exercise this is out of the scope of the present study. All models are driven by publicly available meteorological datasets that have been used and evaluated in numerous studies. As shown in Table 1, two models are using GEOS-5, one model GEOS-FP and one ERA interim meteorology (see in Table 1).

2. Specific comments:

Q: P2, line 1: please add min and max for TFe and LFe deposition fluxes.

A: The minimum and maximum values for the global mean of the deposition fluxes TFe and LFe are added. This part now reads: "The mean global deposition fluxes into the global ocean is here estimated in the range of 10-30 Tg-Fe yr$-1$ and 0.2-0.4 Tg-Fe yr$-1$ for TFe and LFe, corresponding to roughly $\sim$15 Tg-Fe yr$-1$ and $\sim$0.3 Tg-Fe yr$-1$, respectively, for the for the multi model ensemble model mean."

Q: P3, lines 18-22: the fraction of Fe that is bioavailable is still not well known and also depends on phytoplankton species, so I suggest that the authors do not write that labile Fe is a good approximation for bioavailable Fe.

A: We agree with the reviewer that Fe bioavailability is a complex issue and for this we clearly stated this in the manuscript (pp3, lines: 16-22). However to make it more clear we rephrased this part as (please see also our reply to Reviewer 1): "The bioavailability of Fe is a complex issue (e.g., Lis et al., 2015; Morel et al., 2008) and several naming conventions and abbreviations were used to characterise the atmospheric supply of potentially bioavailable Fe to the global ocean (Baker and Croot, 2010; Shi et al., 2012). It has been widely assumed that soluble Fe can be considered, as a first approximation, to be bioavailable (Baker et al., 2006a, 2006b) and a common experimental practice to determine the bioavailable Fe fraction in Fe-containing aerosols is the quantification of Fe in a leachate solution that passes through 0.45 $\mu$m, 0.2 $\mu$m or 0.02 $\mu$m sized filter (see Meskhidze et al., 2016 and ref. therein). However, due to its operational definition, it has been shown that this filterable Fe may contain both the soluble Fe and colloidal forms (Jickells and Spokes, 2001; Raiswell and Canfield, 2012). Upon deposition to the surface ocean, the soluble of Fe delivered through atmospheric pathways can either enter the dissolved Fe pool in the ocean, or precipitate-out as large oxy-hydroxide particles (de Baar and de Jong, 2001; Boyd and Ellwood, 2010; Meskhidze et al., 2017; Turner and Hunter, 2001). Consequently, the impact of atmospheric Fe on marine biogeochemistry depends on both the total Fe (TFe) deposition and its solubility, keeping in mind that the bioavailable fraction of Fe in seawater will then also change due to post-atmospheric deposition ocean processes (e.g., Baker and Croot, 2010; Chen and Siefert, 2004; Meskhidze et al., 2017; Rich and Morel, 1990)."

Q: P4, line 8: "can be also be"

A: Corrected.

Q: P6, lines 2-5: for the role of oxalate on Fe solubility, Paris et al. (2011) could be

cited as well (https://doi.org/10.1016/j.atmosenv.2011.08.068).

A: Reference added.

Q: P7, line 29: please change "in (Albani et al., 2014)" by "in Albani et al. (2014)"

A: Typo corrected.

Q: P18, lines 28-30: "LFe sources are mainly driven by mineral dust aerosols, although a significant fraction (6 to 62%) is due to LFe combustion aerosols, especially over the high-latitudes of the Northern Hemisphere (Ito et al. 2018; companion manuscript to be submitted)." I would rather put this sentence in the previous section.

A: We agree with the reviewer. The sentence has been moved into the previous section.

Q: P19, lines 1-5: the authors compare the seasonal variability of LFe, but I would have liked to see the error bars on Fig. 1, as well as more information on the statistical test (which one was used, P value, n,. . .).

A: Error bars in Fig. 1 are added. For the individual models, however, the results here correspond to one year of simulation. Therefore, no statistics can be derived for the seasonal deposition fluxes which are calculated as the sum of monthly deposition fluxes. We provide further statistics for the ensemble model; the median bias correction factors are presented in Table 3 for each model, together with the lower and upper 95% confidence interval.

Q: Moreover, the authors state that "in most of the cases IMPACT and GEOS-Chem present similar seasonal variation." However, IMPACT is higher in JJA, while GEOS is higher in MAM.

A: We thank the reviewer for pointing this inconsistence. We now corrected this part as: "However, significant differences in the magnitude of the deposition fluxes are calculated between models (Fig. 1). A seasonal maximum in the deposition fluxes is calculated by CAM4 and GEOS-Chem during MAM, attributed to Saharan mineral dust

aerosols, while IMPACT and TM4-ECPL present a seasonal maximum during JJA."

Q: P19, lines 9-11: the authors state that "in the other seasons the 30N maximum is not clearly present", but in JJA, a clear maximum for IMPACT is seen at 30ậŮệN.

A: We now rephrased this part as: "In DJF, and to a lesser extent in JJA, two zonal maxima are shown near the equator and around 30N."

Q: P21, line 12: the authors should explain how they calculate the mean normalized bias. Why would a value of 2.4 mean that the concentrations are underestimated? This is not clear to me.

A: We agree with the reviewer that this sentence is confusing. We have rephrased for clarity as follows: "This reflects that overall the models overestimate TFe surface mass concentrations. However, from Fig. 4 we can see that this overestimate is higher for the highest TFe concentrations near the dust source regions and tend to turn to an underestimate for the lowest concentrations observed over remote oceans." A detailed description of MNB calculations is also added in the manuscript (please see also our reply to Referee#1).

Q: P22, line 3 and Fig. 5: the authors discuss the relationship between Fe solubility and aerosol Fe concentrations but these 2 variables are not independent as the latter one is used to calculate the former one. How do the authors deal with that?

A: This is a good question. We are aware that concentrations and solubility are not independent variables in the calculations since solubility is the ratio of Labile to Total Fe concentrations. At low Fe concentration, for example, Fe solubility of TM4 is similar to ensemble model (red triangles in Fig. 5). This is because the mean Fe solubility is weighted by Fe concentration. More specifically, Fe concentration at low concentration of TM4 is much higher than other models, resulting in similar Fe solubility between TM4 and ensemble model. This indicates that a small number of aerosols with high Fe concentration can determine Fe solubility in bulk samples. We added further work

to address one of the major quantification problems due to the sampling issues in p.25, l.2, "due to short of the sampling frequencies. A comparison of long-term measurements with multi-year hindcast will allow us to assess the model performance to simulate labile Fe concentration under specific events" after "Note, however, that evaluation of monthly mean model results by comparison with the shorter-term (e.g., daily) observations during different sampling periods introduces uncertainties". At the same time, models consider the process of enhancement of Fe solubility. It is therefore interesting to see whether the models are able to capture the fraction of LFe to TFe correctly and this is what we are evaluating in Fig. 5. This figure shows that the models have difficulties to simulate the 4 orders of magnitude variability from 0.02% to 98% in the Fe solubility observed in the atmosphere (Fig. 5a). IMPACT simulates almost 3 orders of magnitude variability in Fe solubility. In the other models including the ensemble model, Fe solubility is less variable (one to two orders of magnitude only). In particular, low solubilities (high concentrations near sources) are overestimated and high solubilities (low concentrations at remote locations) are underestimated. This may indicate that the primary LFe in the models is overestimated and that models are missing solubilisation processes during transport or that those considered in the models are not sufficient effective. The discussion has been added appropriately in the manuscript.

Q: P23, line 12-: How is the lifetime (turnover time) calculated? Is it calculated by dividing the concentration by the deposition flux, both estimated by the model? This could be added in the text.

A: We explain in the caption of Fig. 6 that lifetimes are the calculated atmospheric concentrations (or burdens) divided by total sinks", but for clarity we also added the following explanation in the manuscript: "Figure 6 presents the spatial distribution of TFe lifetime over the ocean (i.e., atmospheric concentrations divided by total sinks), as calculated for the ensemble model."

Q: P24, lines 26-27: please change "similar to what it was pointed out in (Albani et al., 2014) and seen in the dust model intercomparison study of Huneeus et al. (2011)."

By "similar to what was pointed out in Albani et al. (2014) and seen in the dust model intercomparison study of Huneeus et al. (2011)".

A: We corrected the typo: "(Albani et al., 2014)" to "Albani et al. (2014)"

—————————————————————

---

## Author Response (AR1)

Dear editor,

In this document we include a point-by-point response to the reviews with a list of all relevant changes made in the manuscript, and a marked-up manuscript version.
We also include in the manuscript's introduction a more detailed discussion on the need and motivation for this work, as proposed by the editor: "*Overall, the importance of this work lies on an extended review and synthesis of the current knowledge of global atmospheric Fe deposition fluxes in the ocean, aiming to provide ensemble model data to the scientific community, able to be used in ocean biogeochemistry models and as comparative measures for atmospheric models.*" (see p.7, lines 12-15).
We have also added 2 missing co-authors, acknowledgments (see marked-up manuscript), and we have corrected typos. The ensemble model fields from this work are now available online at https://ecpl.chemistry.uoc.gr/GESAMP/.

Kind regards,
S. Myriokefalitakis (on behalf of all co-authors)

We thank the Referee#1 for the careful reading of the paper. Please find below the point-by-point answers to the referee's general and specific comments and technical corrections.

**1. General comments:**

- Q: The authors state that this kind of modelling is the only way to estimate iron deposition (page 4 lines 11ff). But is it really impossible to utilise the large number of observations of Fe concentrations (listed in the SI) to estimate fluxes? I would like to see a better justification for this claim.

  - A: With our statement we do not want to devaluate the importance and the need of observations. We state that models are an excellent way to study atmospheric Fe supply to the oceans and to assess its impacts on a global scale, partly because of episodic nature of atmospheric deposition. Models enable the integration of knowledge and of discontinuous geographically and temporally observations and in synergy with observations are the appropriate tools to study the global spatial and temporal patterns of species such as Fe. We rephrased as: "*The use of global biogeochemical numerical models and surface observations is an excellent way to better understand past, present and future atmospheric supply to the oceans, as well as to quantify the resultant effect on the ocean biological productivity and the carbon uptake.*" (see page 4 lines 15-17).

- Q: The third aim of the work (page 7, top) seems circular – why would future modelling studies find the fluxes calculated in this modelling study useful, other than as comparative measures? I also miss an indication that the work described here is potentially useful in permitting prediction of changes in Fe deposition rates, for example due to anthropogenic activities.

  - A: The 3$^{rd}$ aim refers to the utility of the calculated ensemble Fe deposition as an input for the next-generation of ocean biogeochemistry modelling studies. Our study aims to provide to the scientific community with ensemble TFe and LFe deposition fluxes, as a result of state-of-the-art atmospheric models and satellite retrievals. Currently, most ocean biogeochemistry models use global dust deposition fields to derive the atmospheric Fe input (e.g., Aumont et al., 2015), usually by assuming a constant fraction by mass on dust. Furthermore, to take into account the labile fraction in Fe deposition fluxes, either a constant value is applied or Fe solubility maps from other atmospheric models are used (e.g., Mahowald et al., 2005). Such approaches mean that the ocean data will contain significant assumptions, which the current science no longer supports; e.g., the combustion Fe-containing aerosol and the heterogeneity of soil minerology are neglected as well as the explicitly calculation of the Fe solubilization processes (resulting in lower Fe solubility near the dust source regions and higher Fe solubility over the remote ocean than earlier estimates, bringing closer model results to observations).
  Overall, models need to first be evaluated against observations though to have some level of confidence in their ability and then to use them in order to access for example the anthropogenic effects. We can further use the models for the predicting of changes in Fe deposition rates, especially for the past and the future.

**2. Specific comments:**

- Q: Page 21, line 10 states "*The TFe loading, Fe solubility, and LFe loading from the models are compared with the measurements and presented in Fig. 4.*" This confuses me, since I would use the term loading to mean a flux over time (mass per unit area). The axis labels refer to

concentrations, with units of mass per volume, but the text and the Figure caption use loading. Please clarify.

o A: We changed the term "*loading*" to "*concentrations*" (see page 21 line 26).

- Q: With reference to Figure 4, if I understand correctly (and if I don't then please clarify the text) the MNB values indicate the overall bias of the predictions compared to the data, which would mean that the ensemble model overestimates LFe concentrations by a factor of five. Does it then follow that loadings to the ocean are overestimated by this factor? If so, then the proposed further work doesn't seem to address the issue – Section 5 reads more like a series of minor tweaks than addressing a major quantification problem.

o A: As for the MNB values, overestimates are weighted more than equivalent underestimates. As was noted in p.24, l.24, a similar overestimate in the measured monthly averaged dust concentration from a the short-term cruise measurements was seen in the dust model intercomparison study of Huneeus et al. (2011). The bias may be due to the short duration of the sampling frequencies. We added further work to address one of the major quantification problems in p.25, l.2, "due to to short of the sampling frequencies. *A comparison of long-term measurements with a multi-year modelling will allow assessment of the model performance to capture labile Fe concentrations under specific events*" after "Note, however, that evaluation of monthly mean model results by comparison with the shorter-term (e.g., daily) observations during different sampling periods introduces uncertainties".

We also added the description of MNB in the text as following: "*We use the monthly mean of model output to compare with the measurements. The normalized bias (NB) at a given grid box is calculated as follows:*

$$NB_i = \frac{c_{model,i} - c_{obs,i}}{c_{obs,i}} \qquad (2)$$

*where, $c_{model,i}$ is the modelled aerosol concentration in grid box $i$, and $c_{obs,i}$ is the measured aerosol concentration in the same grid box. When discussing the multi-model results we use the mean of all models, while we also analyze the mean normalized bias (MNB) of the models against measurements (a perfect comparison would have a MNB of 0 and correlation, R, of 1). A model's MNB is derived as the arithmetic mean of all $NB_i$ values, thus overestimates are weighted more than equivalent underestimates.*" (see page 21 lines 27-28 and page 22 lines 1-7).

- Q: Evaluating the importance of atmospherically deposited Fe depends greatly upon assessing the fate of the metal in ocean water. According to the authors "Upon deposition to the surface ocean, this fraction of Fe from the atmosphere can either enter the dissolved Fe pool, or precipitate-out as large oxy-hydroxide particles (Meskhidze et al., 2017)". I am surprised that the cited study, which worked with high Fe concentrations and did not explore the influence of light on iron chemistry, is considered to represent the state of knowledge in this area. I am also surprised that neither this reference nor the paper under review cites the book by Turner and Hudson "The Biogeochemistry of Iron in Seawater" (Wiley 2001).

A: The study by Meskhidze et al. (2017) was designed to represent the processes affecting the soluble Fe deposited to the open oceans through atmospheric pathways on a time scale of sec to minutes. The concentrations were selected to be representative of wet removal (i.e., rainout and washout) which is the dominant removal mechanism over the remote oceans. The reference is used here, because, as far as we know, this is the first study that explored the role of atmospheric organic ligands on Fe solubility after deposition to the surface ocean. However, we agree with the reviewer tha additional references, particularly on the effects of oceanic ligands and

photochemistry need to be provided. The revised manuscript now reads: "*Upon deposition to the surface ocean, the soluble of Fe delivered through atmospheric pathways can either enter the dissolved Fe pool or precipitate-out as large oxy-hydroxide particles (de Baar and de Jong, 2001; Boyd and Ellwood, 2010; Meskhidze et al., 2017; Turner and Hunter, 2001).*" (see page 3 lines 27-29).

- Q: Is there any prospect of using the Fe loadings reported here to simulate Fe concentrations in the ocean? I realize that this may be outside the scope of the present paper, but some indication of possibilities would be welcome.

  - A: We included the following part in the conclusions: "*Although the calculation of the Fe concentrations in the ocean is outside the scope of this paper, we expect that the Fe deposition fluxes here provided will be used in oceanic models.*" (see page 27 lines 7-9).

- Q: It is not clear to me whether FeD deposited to the ocean is considered "inert" or whether it can yield significant dissolved. Maybe this could be explained. If it is not considered to be a source, then it is not so important to get the global fluxes correct, and the focus should be on the LFe.

  - A: Total Fe deposited in the ocean is important for the assessment of the fate of Fe in the ocean. Total Fe is needed for the comparison of particulate Fe with the measurements in the ocean biogeochemistry models (e.g., Ye and Völker, 2017). Additionally, less labile Fe in total Fe may be potentially utilized by marine organisms. Note that ocean biogeochemistry models (e.g., Aumont et al., 2015) take into account both the total and the soluble deposited Fe for chemistry calculations, assuming some fractions of less labile Fe in total Fe are dissolved in the ocean. For example, Aumont et al. (2015) considers that the particulate Fe from dust experiences dissolution in the water column, with the dissolution rate computed assuming that during sinking of mineral particles particulate Fe dissolves by about 0.01% per day (Bonnet, 2004). Therefore, both the total and the labile Fe deposition fluxes are needed. However, the dissolution of Fe from FeD is species depended and affected by spatiotemporal variations in the ocean. The following explanation has been added in the text in the introduction: "*Both the TFe and LFe atmospheric deposition can be used in ocean biogeochemical modelling. For example, total Fe is needed for comparisons of particulate Fe with the measurements in the ocean biogeochemistry models (e.g., Ye and Völker, 2017), while LFe can be assumed as readily available to the marine ecosystem. Note that the less labile fraction of Fe in TFe can be slowly dissolved from particulate Fe in the ocean during sinking of mineral particles (e.g., roughly 0.01% per day; Bonnet, 2004), with the dissolution of Fe, however being species depended and affected by spatiotemporal variations in the ocean.*" (see page 4 lines 26-31).

- Q: As I understand it, a similar loading (to LFe) of dissolved Fe to the oceans comes from rivers. Could the authors briefly explain why this is not considered as important as the atmospherically-deposited form?

  - A: According to the recent study of Tagliabue et al. (2016), riverine inputs are considering 1-2 order of magnitudes smaller than the atmospheric dust deposition to the global ocean. We inclued the following part in the manuscript (introduction), to further refer to the other known sources of Fe in the global ocean: "*However, significant Fe inputs from continental margins and hydrothermal vents are also supplied to the global*

*ocean, regulating the ocean biogeochemical cycles. Moreover, riverine Fe inputs are currently estimated 1-2 orders of magnitudes smaller that the atmospheric pathway (e.g., Tagliabue et al., 2016), affecting mainly coastal regions, while icebergs and glaciers could also be important to the polar oceans (Raiswell et al., 2016).*" (see page 3 lines 12-16).

- Q: Section 2.1.2 introduces the presence of oxalate in aerosols, without explanation of its sources and why other carboxylic acids are not considered. I am not at all expert in this area, it appears as though oxalate is assumed or known to be dominant – if so then its strong solubilising properties are clearly important. I would appreciate some references to justify the assumption that oxalate is truly dominant in governing aerosol Fe solubility.

  - A: Indeed, numerous organic compounds, such as acetate, formate, oxalate, malonate, succinate, glutarate, glycolate, lactate, tartrate and humic like substances (HULIS) can be found in atmospheric waters. However, oxalate, malonate, tartrate and humic acid have been observed to enhance Fe solubility (e.g., Paris et al., 2010, 2011). For all these organic ligands, positive dependences of iron solubility to organic concentrations were observed and revealed that the extent of organic complexation on iron solubility decreased in the following order: oxalate > malonate = tartrate > humic acid (Paris et al., 2011). Therefore, this study confirmed that among the known atmospheric organic binding ligands of Fe, oxalate is the most effective ligand in promoting dust iron solubility under atmospheric conditions. Furthermore, observations in the atmosphere, point to oxalate as the most abundant organic ligand (e.g., Kawamura and Ikushima, 1993; Kawamura and Sakaguchi, 1999). Oxalate originates from multiphase chemistry of organics, but has also weak anthropogenic primary sources (see Myriokefalitakis et al., 2011 and Lin et al., 2014 for a comprehensive global modelling study of atmospheric oxalate). Therefore, atmospheric models use oxalate to study the effect of organic ligands on Fe dissolution. However, the lack of experimental data for Fe-containing minerals mixed with a variety of organic ligands in solution is an important source of uncertainty. For clarity, in Sect. 2.1.2 we added after the first sentence the following explanatory text: "*Oxalate is, however, used in models as a proxy of all organic ligands for ligand-promoted dissolution since 1) it is the most abundant in the atmosphere (e.g., Kawamura and Ikushima, 1993; Kawamura and Sakaguchi, 1999) originating mainly from secondary sources and only a weak contribution from combustion primary sources (e.g., Myriokefalitakis et al., 2011) and 2) it is the most effective ligand in promoting iron solubilisation (e.g., Paris et al., 2011). We note, however, that more work is required to elucidate the role of other ligands that may promote Fe dissolution in future studies.*" (see page 12 lines 3-9).

- Q: The right-hand maps in Figure S4 are not informative. Is it possible – or do the authors consider it worthwhile? – to show primary sources of LFe?

  - A: As we state in the text, not all the models simulate the LFe primary and secondary sources in the same manner of dust and combustion aerosols. For a fairer comparison in Fig. S4, we show the primary (i.e., emissions) and secondary (i.e., atmospheric processing) sources together. As we state in the manuscript "*The models use significantly different assumptions to describe the total LFe source to the atmosphere and therefore primary (emissions) and secondary (atmospheric processing) sources cannot be accurately separated from rapid formation assumed in coarse-scale models.*" (see page 17 lines 9-12). A detailed description of models' parameterizations as well as the differences among them with regard to the LFe sources, are also presented in Sect. 2.1.

**3. Technical Corrections:**

- Q: Page 5 line 15 This sentence needs improvement.

  - A: We rephrased the text between lines 13-19 as follows: "*During atmospheric transport, coating of Fe-containing dust particles by acidic compounds (e.g., sulfates and nitrates) increases the Fe solubility. When this process is taken into account in model simulations (e.g., Meskhidze et al., 2005) it aids in explaining the observations. Indeed, measurements of the fresh dust particles present low (<<1%) initial solubilities (Chuang et al., 2005; Fung et al., 2000; Hand et al., 2004; Sedwick et al., 2007), while high aerosol solubilities are commonly observed at lower dust concentrations far from sources (Baker and Jickells, 2006; Sholkovitz et al., 2012; Oakes et al., 2012). Atmospheric processing of dust (Kumar et al., 2010; Meskhidze et al., 2003; Srinivas et al., 2014) is considered as the best candidate to explain these observations.*" (see page 5 lines 21-28).

- Q: Table S3. What does "NaN" mean? (not analysed I guess, but please say).

  - A: NaN is replaced with "-", which means that data are not available. We have also modified and explain it now in the Table S3 caption.

- Q: Figure S7 should have "continuous" not "*continues*".

  A: Done

- Q: Page 21 line 19. I assume that SH = southern hemisphere? Is this so very well known?

  A: We replaced "SH" with "*the Southern Hemisphere*".

- Q: Page 21 line 22. Should it read 0.50-0.56?

  - A: The value is correct. The differences in Fe solubility trend between CAM4 and TM4-ECPL can be partly seen from Fig. 5.

- Q: My main concern is about the interannual variability of the models. As stated in Table 1, the simulated years are different for each model but the interannual variability of each model is not presented nor discussed.

  - A: This work uses single year model results only. Therefore, no year-to-year variability is possible to be presented in our post-processed analysis for the present atmosphere. However, our analysis shows the variability derived from the structural differences between models.

- Q: Moreover, no requests for meteorological conditions or emission inventories have been set to the model simulations and the sensitivity of each model to these parameters are also not discussed.

  - A: The aim of this work is to describe the current state of Fe global atmospheric deposition modeling and provide a multi-model ensemble of Fe atmospheric deposition fluxes of the current estimates, characterized with regard to observations (see introduction). It aims also to understand the origin of the respective model differences over the oceanic regions among the participating models and not to conclude which model is the best. As discussed in the model description (section 2), the participating models use different parameterizations to simulate the Fe-cycle in the atmosphere. Although using the same meteorology and the same emission inventories in the participating models would have been a very interesting exercise this is out of the scope of the present study. All models are driven by publicly available meteorological datasets that have been used and evaluated in numerous studies. As shown in Table 1, two models are using GEOS-5, one model GEOS-FP and one ERA interim meteorology (see in Table 1).

**2. Specific comments:**

- Q: P2, line 1: please add min and max for TFe and LFe deposition fluxes.

  - A: The minimum and maximum values for the global mean of the deposition fluxes TFe and LFe are added. This part now reads: "*The mean global deposition fluxes into the global ocean is here estimated in the range of 10-30 Tg-Fe yr$^{-1}$ and 0.2-0.4 Tg-Fe yr$^{-1}$ for TFe and LFe, corresponding to roughly ~15 Tg-Fe yr$^{-1}$ and ~0.3 Tg-Fe yr$^{-1}$, respectively, for the for the multi model ensemble model mean.*" (see page 2 lines 3-6).

- Q: P3, lines 18-22: the fraction of Fe that is bioavailable is still not well known and also depends on phytoplankton species, so I suggest that the authors do not write that labile Fe is a good approximation for bioavailable Fe.

  - A: We agree with the reviewer that Fe bioavailability is a complex issue and for this we clearly stated this in the manuscript (pp3, lines: 16-22). However to make it more clear we rephrased this part as (please see also our reply to Reviewer 1): "*The bioavailability of Fe is a complex issue (e.g., Lis et al., 2015; Morel et al., 2008) and several naming conventions and abbreviations were used to characterize the*

*atmospheric supply of potentially bioavailable Fe to the global ocean (Baker and Croot, 2010; Shi et al., 2012). It has been widely assumed that soluble Fe can be considered, as a first approximation, to be bioavailable (Baker et al., 2006a, 2006b) and a common experimental practice to determine the bioavailable Fe fraction in Fe-containing aerosols is the quantification of Fe in a leachate solution that passes through 0.45 μm, 0.2 μm or 0.02 μm sized filter (see Meskhidze et al., 2016 and ref. therein). However, due to its operational definition, it has been shown that this filterable Fe may contain both the soluble Fe and colloidal forms (Jickells and Spokes, 2001; Raiswell and Canfield, 2012). Upon deposition to the surface ocean, the soluble of Fe delivered through atmospheric pathways can either enter the dissolved Fe pool in the ocean, or precipitate-out as large oxy-hydroxide particles (de Baar and de Jong, 2001; Boyd and Ellwood, 2010; Meskhidze et al., 2017; Turner and Hunter, 2001). Consequently, the impact of atmospheric Fe on marine biogeochemistry depends on both the total Fe (TFe) deposition and its solubility, keeping in mind that the bioavailable fraction of Fe in seawater will then also change due to post-atmospheric deposition ocean processes (e.g., Baker and Croot, 2010; Chen and Siefert, 2004; Meskhidze et al., 2017; Rich and Morel, 1990)."* (see page 3 lines 19-30 and page 4 lines 1-3).

   o

- Q: P4, line 8: "can be also be"

   o A: Corrected.

- Q: P6, lines 2-5: for the role of oxalate on Fe solubility, Paris et al. (2011) could be cited as well (https://doi.org/10.1016/j.atmosenv.2011.08.068) .

   o A: Reference added (see page 6 line 13).

- Q: P7, line 29: please change "in (Albani et al., 2014)" by "in Albani et al. (2014)"
   o A: Typo corrected.

- Q: P18, lines 28-30: "LFe sources are mainly driven by mineral dust aerosols, although a significant fraction (6 to 62%) is due to LFe combustion aerosols, especially over the high-latitudes of the Northern Hemisphere (Ito et al. 2018; companion manuscript to be submitted)." I would rather put this sentence in the previous section.

   o A: We agree with the reviewer. The sentence has been moved into the previous section.

- Q: P19, lines 1-5: the authors compare the seasonal variability of LFe, but I would have liked to see the error bars on Fig. 1, as well as more information on the statistical test (which one was used, P value, n,. . .).

   o A: Error bars in Fig. 1 are added. For the individual models, however, the results here correspond to one year of simulation. Therefore, no statistics can be derived for the seasonal deposition fluxes which are calculated as the sum of monthly deposition fluxes. We provide further statistics for the ensemble model; the median bias correction factors are presented in Table 3 for each model, together with the lower and upper 95% confidence interval.

- Q: Moreover, the authors state that "in most of the cases IMPACT and GEOS-Chem present similar seasonal variation." However, IMPACT is higher in JJA, while GEOS is higher in MAM.

    - A: We thank the reviewer for pointing this inconsistence. We now corrected this part as: *"However, significant differences in the magnitude of the deposition fluxes are calculated between models (Fig. 1). A seasonal maximum in the deposition fluxes is calculated by CAM4 and GEOS-Chem during MAM, attributed to Saharan mineral dust aerosols, while IMPACT and TM4-ECPL present a seasonal maximum during JJA."* (see page 9 lines 16-19).

- Q: P19, lines 9-11: the authors state that "in the other seasons the 30N maximum is not clearly present", but in JJA, a clear maximum for IMPACT is seen at 30◦N.

    - A: We now rephrased this part as: *"In DJF, and to a lesser extent in JJA, two zonal maxima are shown near the equator and around 30N."* (see page 19 lines 22-23).

- Q: P21, line 12: the authors should explain how they calculate the mean normalized bias. Why would a value of 2.4 mean that the concentrations are underestimated? This is not clear to me.

    - A: We agree with the reviewer that this sentence is confusing. We have rephrased for clarity as follows: *"This reflects that overall the models overestimate TFe surface mass concentrations. However, from Fig. 4 we can see that this overestimate is higher for the highest TFe concentrations near the dust source regions and tend to turn to an underestimate for the lowest concentrations observed over remote oceans."* (see page 22 lines 10-12).
    A detailed description of MNB calculations has been also added in the manuscript (please see Eq. 2, page 22, line 2 and our reply to Referee#1).

- Q: P22, line 3 and Fig. 5: the authors discuss the relationship between Fe solubility and aerosol Fe concentrations but these 2 variables are not independent as the latter one is used to calculate the former one. How do the authors deal with that?

    - A: This is a good question. We are aware that concentrations and solubility are not independent variables in the calculations since solubility is the ratio of Labile to Total Fe concentrations. At low Fe concentration, for example, Fe solubility of TM4 is similar to ensemble model (red triangles in Fig. 5). This is because the mean Fe solubility is weighted by Fe concentration. More specifically, Fe concentration at low concentration of TM4 is much higher than other models, resulting in similar Fe solubility between TM4 and ensemble model. This indicates that a small number of aerosols with high Fe concentration can determine Fe solubility in bulk samples. We added further work to address one of the major quantification problems due to the sampling issues: *"Note, however, that evaluation of monthly mean model results by comparison with the shorter-term (e.g., daily) observations during different sampling periods introduces uncertainties, due to short of the sampling frequencies; a comparison of long-term measurements with a multi-year modelling will allow assessment of the model performance to capture labile Fe concentrations under specific events."* (page 26, lines 4-9).

    - At the same time, models consider the process of enhancement of Fe solubility. It is therefore interesting to see whether the models are able to capture the fraction of LFe

to TFe correctly and this is what we are evaluating in Fig. 5. This figure shows that the models have difficulties to simulate the 4 orders of magnitude variability from 0.02% to 98% in the Fe solubility observed in the atmosphere (Fig. 5a). IMPACT simulates almost 3 orders of magnitude variability in Fe solubility. In the other models including the ensemble model, Fe solubility is less variable (one to two orders of magnitude only). In particular, low solubilities (high concentrations near sources) are overestimated and high solubilities (low concentrations at remote locations) are underestimated. This may indicate that the primary LFe in the models is overestimated and that models are missing solubilisation processes during transport or that those considered in the models are not sufficient effective. The discussion has been added appropriately in the manuscript.

- Q: P23, line 12-: How is the lifetime (turnover time) calculated? Is it calculated by dividing the concentration by the deposition flux, both estimated by the model? This could be added in the text.

    o A: We explain in the caption of Fig. 6 that lifetimes are the calculated atmospheric concentrations (or burdens) divided by total sinks", but for clarity we also added the following explanation in the manuscript:

[revised manuscript text omitted]